# Recent Developments and Future Perspective on Electrochemical Glucose Sensors Based on 2D Materials

**DOI:** 10.3390/bios12070467

**Published:** 2022-06-28

**Authors:** Sithara Radhakrishnan, Seetha Lakshmy, Shilpa Santhosh, Nandakumar Kalarikkal, Brahmananda Chakraborty, Chandra Sekhar Rout

**Affiliations:** 1Centre for Nano and Material Science, Jain University, Jain Global Campus, Jakkasandra, Ramanagara, Bangalore 562 112, Karnataka, India; sitharark1994@gmail.com; 2International and Inter University Centre for Nanoscience and Nanotechnology, Mahatma Gandhi University, Kottayam 686 560, Kerala, India; seethalakshmy2015@mgu.ac.in (S.L.); shilpas2296@mgu.ac.in (S.S.); nkkalarikkal@mgu.ac.in (N.K.); 3School of Pure and Applied Physics, Mahatma Gandhi University, Kottayam 686 560, Kerala, India; 4School of Nanoscience and Nanotechnology, Mahatma Gandhi University, Kottayam 686 560, Kerala, India; 5High Pressure and Synchroton Radiation Physics Division, Bhabha Atomic Research Centre, Trombay, Mumbai 400 085, Maharashtra, India; 6Homi Bhabha National Institute, Mumbai 400 094, Maharashtra, India

**Keywords:** glucose sensor, 2D materials, electrochemical sensor, wearable and flexible sensor, photoelectrochemical sensors

## Abstract

Diabetes is a health disorder that necessitates constant blood glucose monitoring. The industry is always interested in creating novel glucose sensor devices because of the great demand for low-cost, quick, and precise means of monitoring blood glucose levels. Electrochemical glucose sensors, among others, have been developed and are now frequently used in clinical research. Nonetheless, despite the substantial obstacles, these electrochemical glucose sensors face numerous challenges. Because of their excellent stability, vast surface area, and low cost, various types of 2D materials have been employed to produce enzymatic and nonenzymatic glucose sensing applications. This review article looks at both enzymatic and nonenzymatic glucose sensors made from 2D materials. On the other hand, we concentrated on discussing the complexities of many significant papers addressing the construction of sensors and the usage of prepared sensors so that readers might grasp the concepts underlying such devices and related detection strategies. We also discuss several tuning approaches for improving electrochemical glucose sensor performance, as well as current breakthroughs and future plans in wearable and flexible electrochemical glucose sensors based on 2D materials as well as photoelectrochemical sensors.

## 1. Introduction

Diabetes was the seventh leading cause of mortality worldwide in 2016, according to the World Health Organization (WHO). Diabetes-related early death has been continuously rising. There were around 1.6 million deaths globally in 2019. According to the WHO, diabetes is a major cause of various health effects, including blindness, lethargy, kidney failure, strokes, numbness, and heart attacks [1]. Our mental pictures of diabetes are of overweight people and the elderly. According to current ICMR studies, over 96,000 children suffer from diabetes, with nearly 16,000 new cases diagnosed each year in children under the age of 14. As a result, there is a risk that diagnoses may be overlooked and treatment will be delayed, resulting in a potentially fatal illness [2]. Thus, glucose monitoring is an important tool for diabetes diagnosis [3].

Several types of biosensors have been developed for this purpose. They include invasive, non-invasive glucose monitoring systems, and continuous glucose monitoring systems (CGM). In this sector, reliable and accurate glucose monitoring is always a major difficulty. Recently, a report by Nava et al. provided a systematic review of the comparative diagnostic accuracy of CGM devices in preterm infants. They used a bivariate model to summarize the receiver operating characteristic curve (ROC) curve and extract the area under the curve (AUC). According to their findings, CGM systems have low sensitivity for detecting hypoglycemia in preterm newborns but good accuracy for identifying hyperglycemia [4]. As a result, new approaches are continually being developed, and minimally invasive glucose monitoring technologies have sparked a lot of interest in this field. A biosensor’s main components include:(1)A **biological element** that distinguish the analyte from other substances;(2)A **transducer**, which converts the biorecognition event into a quantifiable signal;(3)A **processing system** that converts a quantifiable signal into a readable signal [5].

Various types of transducers employed for the detection are calorimetric, thermometric, magnetic, optical, piezoelectric, electrochemical transducers, etc. Calorimetric transducers rely on changes in heat energy caused by chemical reactions between the bio-analyte and enzymes [6]. Thermistors are commonly used to detect temperature changes. Magnetic transducers detect changes in the magnetic properties of magnetic fields such as direction, intensity, and flux [7]. Optical transducers make use of the interaction of the optical field with bio components [8]. Piezoelectric transducers operate on the basis of a change in the frequency of oscillations that is proportional to the mass bound to the surface of the electrode [9]. Finally, electrochemical transducers rely on changes in electrical signals caused by electrochemical processes at the electrode surface, which are directly proportional to analyte concentration [5]. Among all these, electrochemical glucose sensors have a high potential due to several appealing features such as low cost, fast response time, high sensitivity, ease of operation, and excellent miniaturization and construction potential for portable glucose sensing devices for point-of-care (POC) applications [10]. The advantage and disadvantages are given below in Table 1.

The transduction techniques used in electrochemical glucose sensors are amperometry, conductometry; capacitive, impedance spectroscopy; voltammetry and potentiometry (ion-selective) field, and effect. Its features are discussed below in Table 2.

With the great advancements in nanotechnology over the last decade, there has been a surge in research interest in nanomaterial-based electrochemical glucose sensors. Combining glucose biosensors with the distinct properties of developing nanostructures based on two-dimensional (2D) advanced materials has resulted in significantly better sensor performance in terms of sensitivity and selectivity [15]. These recent advances have also improved the key sensor properties related to biofouling, long-term use, limit of detection, and wearability.

Reviews on electrochemical glucose sensors have already been published, encompassing their operational principles [16], history [17], important breakthroughs, and the issues these electrochemical glucose sensors confront. There are also papers discussing the utilization of nanostructures for electrochemical glucose sensors [10]. For example, graphene-based electrochemical sensors for the detection of glucose were well-reviewed by Zhang et al. [18]. After graphene, a large number of 2D materials were discovered and are also widely used in electrochemical glucose sensing. However, comprehensive reviews discussing the 2D materials for enzymatic and non-enzymatic electrochemical sensors are rare.

The first half of this review presents the mechanisms of electrochemical glucose sensors, followed by a discussion of recent advancements based on the use of 2D materials to improve glucose sensor performance. The second section focuses on wearable glucose sensors and provides a comprehensive analysis of different innovative non-invasive detection concepts, as well as associated limitations, challenges, and prospects. The fast advancement of wearable and mobile technologies has sparked considerable interest among researchers in non-invasive glucose biosensing. The last decade has also witnessed the development of photoelectrochemical glucose sensors. Despite its late beginnings, PEC glucose sensing has grown rapidly as a significant branch of electrochemical glucose detection. There are some noteworthy reviews available that state advancements in PEC sensing and performance. However, no review concentrating on PEC glucose sensing based on 2D materials has been described thus far.

To further understand the achievements of the last decade in depth, the readers must go through recent topics such as nanomaterials-based electrochemical glucose sensors, graphene-based electrochemical glucose sensors, PEC sensors, advances in wearable sensors, and so on [1,11,19,20].

This review covers the state of the art of electrochemical glucose sensors based on 2D materials developed primarily from 2010 onwards. Figure 1 shows the summary of the review.

### Commercialy Available Electrochemical Glucose Sensors

The Graphene Market and 2D Materials Assessment Report (2021–2031) predicted 18 primary application areas and the transition of graphene biosensors from the laboratory to the commercial market in the coming decades. According to this assessment analysis, the graphene market is expected to expand from $100 million in 2020 to $700 million by 2031 [21]. This success will take time and will be the result of ongoing global research and commercialization initiatives. There is a race to commercialize graphene-based glucose sensors, and their introduction will transform the way diseases are diagnosed and treated. Patients will obtain test results quickly, diagnoses will arrive on-site in minutes, and treatment can begin immediately, saving important and precious time. For example, in partnership with integrated graphene, researchers from the University of Bath recently developed an electrochemical sensor whose function is unaffected by changes in pH or temperature. Boronic acid is used in the new sensor, which is connected to a graphene foam surface. On top, an electroactive polymer layer bind to the boronic acid. When glucose is present, it binds to the boronic acid competitively, displacing the polymer. The sensor generates an electric current proportional to the amount of polymer displaced, allowing the concentration of glucose in the sample to be precisely detected [22]. Even though this 2D-material-based glucose sensor shows quite satisfactory properties for glucose sensing, it should be noted that the electrochemical glucose sensors in the current market are still based on noble metal catalysts. However, there was a trend for graphene-based electrochemical glucose sensors. The global market trends and opportunities show that glucose sensors based on 2D materials are on an upward trend.

## 2. Electrochemical Glucose Sensors

### 2.1. Glucose Sensing Mechanism

Electrochemical biosensors are the most often used sensors today due to their improved sensitivity, repeatability, ease of maintenance, and low cost. Electrochemical glucose sensors are further classified as enzyme-based or non-enzyme-based. The non-enzymatic/enzymatic techniques are distinguished by the direct/indirect electrooxidation reaction of glucose molecules in the whole blood [23].

Glucose readings are frequently based on interactions with one of three enzymes: hexokinase, glucose oxidase (GO_x_), or glucose-1-dehydrogenase (GDH), with GO_x_ and GDH interactions assisting in glucose self-monitoring biosensing.

For the electrochemical sensing of glucose, three detection approaches are available: detecting the amount of consumed oxygen, the amount of hydrogen peroxide produced by enzyme activity, or using a diffusible or immobilized mediator to transport electrons from the GOx to the electrode. GDH-pyrroloquinolinequinone (PQQ) and GDH-nicotinamide-adenine dinucleotide (NAD) are two frequently utilised GDH-based amperometric biosensor families [24]. The following sections will go over the various mechanisms involved in enzymatic and non-enzymatic glucose sensing.

#### 2.1.1. The Sensing Mechanism of the Enzymatic Glucose Sensor

The history of the original glucose-sensing technology begins in 1962, with Clarks and Lyons’ description of the first enzyme-based electrode [25]. The first electrode’s basic premise was the oxidation of D-glucose by molecular oxygen by glucose oxidase entrapped on a semipermeable dialysis membrane supported by an oxygen electrode. Immobilized GOx promotes the oxidation of glucose molecules, forming gluconic acid and hydrogen peroxide. The redox cofactor flavin adenine dinucleotide (FAD), which reduces to FADH_2_ upon electron acceptance, contributes to GO_X_’s catalytic activity. H_2_O_2_ is formed as a result of cofactor regeneration. Hydrogen peroxide is oxidized at a catalytic, traditionally platinum (Pt) anode that counts the number of electron transfers as a function of the number of glucose molecules in the blood.

Clark patented his subsequent investigation with two electrode systems to monitor glucose sensing in 1970. Electro-inactive materials were transformed into electroactive species in the proposed system using enzyme-based sensing. The enzymes between the membrane and the first electrode transformed the substrate into an electroactive state despite the presence of interfering species in the sample. Still, the second electrode was extremely sensitive to these interfering species in the sample—the difference in currents allowed for successful glucose detection. The initial glucose sensor, however, was not a biosensor. The reflectometer served as the foundation for the early generations of all glucose-sensing devices. The Ames Reflectance Meter (ARM), launched in 1961, was never user-friendly or cost-effective. Clark’s work was moved to Yellow Springs to develop the first-generation commercial glucose sensor. The first commercial glucose sensor, introduced in 1975, was enzyme-based with an amperometric measurement of H_2_O_2_ from a human blood sample by the Yellow Springs Instrument Company (as Model 23A YSI analyzer) (Figure 2) [26,27].

With advancements in surface chemistry and device fabrication, the field of commercial electrochemical glucose biosensors has been advanced [28]. Many inorganic redox couples, mediators such as ferricyanide, ferrocene, tetrathiafulvalene (TTF), quinines, thionine tetracyanoquinodimethane (TCNQ), methyl viologen, and methylene blue, and organic colors, cater electrons from the enzyme layer to the working electrode surface [24]. Because the reaction with the enzyme is fast, ferrocene is an often-employed mediator. The mediator can function in a wide range of pH settings and has reversible electron kinetics. Second-generation glucose sensors outperformed first-generation glucose sensors from 1984 because the redox couple or mediator-based approach was generally quite sensitive to dissolved oxygen even at lower potentials and extreme interference ambience. ExacTech by Medisense Inc. was the first home-based self-monitoring glucose sensor to hit the market, following the discovery of a screen-printing process that enabled the creation of wearable, disposable, compact, and pocket-friendly electrodes. Each strip consisted of miniaturized screen-printed reference and working electrodes, with the working electrode coated with the important sensing components, including an electron-shuttle redox mediator, glucose oxidase linking agent and stabilizer [29]. The relevance of second-generation glucose detectors is still being debated; however, with the introduction of finger pricking home-based second-generation glucose sensors, the development of glucose meters accelerated tremendously.

Even the revolution of home-based diabetic-patient-friendly glucose metres using mediated systems did not contribute significantly to the nature of the biochemical structure of the GOx enzyme, the relative toxicity and solubility of the intermediates, and the general poor stability of these under varying pH, temperature, and humidity conditions. As a result, scientists all over the world have been working hard to improve performance by employing techniques that facilitate electron transfer between the GOx redox center and the electrode surface, such as the chemical modification of GOx with electron-relay groups, enzyme wiring of GOx by electron-conducting redox hydrogels, and the use of nanomaterials as electrical connectors, among others. Another dependable option, however, was the direct transmission of electrons from the enzyme sites to the electrode without the presence of intermediaries. Such needle-type implantable devices can be used for continuous glucose monitoring at low potentials near the enzyme’s redox potential. The absence of mediators allows the biosensors to pick glucose molecules better [24].

On the other hand, third-generation sensors are still in the early stages of development. However, those built on nano-mesoporous electrode surfaces hold promise but have substantial limitations. Only a few enzymes have been shown to transport electrons directly from the enzyme to the working electrode. However, these published papers on glucose detection do not provide sufficient evidence to explicate mediator-free glucose detection. Electrode-enzyme detection should be tuned for proper electron transfer. Carbon materials, including single-walled and multi-walled carbon tubes incorporated with nitrogen and phosphorous species, are better platforms for achieving high-performance biosensors for detecting glucose. The nature and amount of the nitrogen and phosphorus species combined in the carbon material surface can be selectively controlled. These species act as anchoring groups for the immobilization of the PQQ-GDH [30,31,32].

#### 2.1.2. The Sensing Mechanism of Non-Enzymatic Glucose Sensors

As previously stated, enzymatic glucose sensors have certain inherent limitations, such as they require appropriate system storage, enzyme denaturation resulting from the change in environment, expensive preparation, protease digestion, time-consuming purification process, thermo-chemical deformation, high cost, poor reproducibility caused by extraneous oxygen sources, lack of stability in harsh conditions, which makes system sterilization difficult, and monotonous enzyme immobilization. The decreased performance produced by the immobilization process remains a barrier that must be overcome [33,34]. Nanomaterial-assisted electrochemical processes via non-enzymatic glucose sensing can fully describe the shortcomings of enzyme-based glucose sensing. As a result, non-enzymatic glucose sensors can be thought of as next-generation glucose detection meters. Most enzyme-free sensors are more sensitive than enzyme-based glucose detectors [35].

Non-enzymatic glucose sensors have a wide range of applications, including catalysis, biomedical devices, and environmental monitoring [36,37,38,39]. Since noble metals and related metal alloys have strong catalytic activity for non-enzymatic glucose electrooxidation, they have focused primarily on their applications [40]. There are currently two major approaches to improving the performance of enzyme-free glucose sensors: one is to change the topological structure of the material to increase the specific surface and electrochemically active sites, and the other is to form hybrids by combining or doping with other materials. Metals and their compounds, metal-organic frameworks (MOFs), conductive polymers, inorganic carbon materials and other materials used to construct glucose sensors form the sensor’s working electrode. This composite of two or more materials fully exploits their respective advantages in building glucose non-enzyme sensors, thereby boosting non-enzyme sensor performance, such as higher sensitivity, wider linear range, and lower detection limit.

Even though platinum nanoparticles are widely used in a non-enzymatic glucose sensor, there are still significant obstacles to overcome, such as poor selectivity, slow kinetics, and limited sensitivity. When platinum is changed with Co, Au, Ni, and Fe alloys and adorned with MWCNTs/carbon, the surface catalytic activity changes to improve stability and sensitivity. A specific synthesis method has been used to address the limitations of Pd BNP as well as for sensing glucose by considering effective combinations such as MOS_2,_ CNTs, and so on [35]. However, noble metals are generally expensive, and the use of non-noble metals is always appreciated. As a result, non-enzymatic electro-oxidation of glucose has been widely investigated utilizing a variety of nanomaterials ranging from noble metals to non-noble metals such as Cu, Zn, Co, Mn, Fe, and Ni, as well as metal sulfides and oxides.

Increasing the exposed active surface of a non-enzymatic glucose sensor is an effective routine for improving analytical performance. As a result, many materials with favorable porosity (microporous, mesoporous, and even macroporous) are expected to find widespread use in the fabrication of satisfactory glucose sensing devices. Because of their better conductivity, high surface area, and improved electron transfer, nanomaterials supported on various substrates, including graphene and its derivatives-based substrates, can improve glucose electrooxidation and detection. A review published in 2020 summarizes the performance of 2D noble and non-noble materials pretty well [40].

Two-dimensional (2D) materials’ promising physicochemical features show them to be an emerging class of possibilities for application in sensing devices. They can also be applied to the IoT platform due to their high surface-area-to-volume ratio, leading to great sensitivity to target glucose molecules (Figure 3) [41]. Furthermore, due to their high mechanical strength, flexibility, and optical transparency, sensors based on 2D materials can also be miniaturized.

Physisorption and chemisorption are the two unique methods by which 2D materials interact with target molecules/ions. Physisorption is caused by molecules/ions interaction with 2D materials’ surface in the absence of covalent bonding, whereas chemisorption is caused by covalent bonding between molecules/ions and the surface of 2D materials. Non-covalent bonding, on the other hand, is favored for quick response. Because of its honeycomb structure, graphene material prefers to establish pi-bonds with molecules [42] rather than the van Der Waals force between MoS_2_ and molecules, which is indicative of TMDs [43]. Graphene’s thin single layer promotes high carrier layer mobility, making it an excellent 2D material for glucose sensing. The zero-band gap of the graphene substrate is responsible for the hazard-free charge flow [44]. In the case of glucose detection, however, decorating graphene, building a composite with other materials, or utilizing functionalized GO or rGO improve the sensitivity of graphene. N-doped graphene was produced by treating graphene with nitrogen plasma, while Cu-doped graphene was produced via electrodeposition that was adjusted by manipulating the deposition duration and voltage [45,46]. Both procedures produced repeated stable, sensitive, and selective results. Glucose sensing is catalyzed by GOx. However, GO is incompatible with organic polymers. The polymer N-succinimidyl acrylate is used to chemically modify GO with GOx. Figure 4 depicts the effect of several graphene electrodes employed under varying pH settings in an interfering environment [47].

Electrochemical etching, nano-band electrode fabrication, and the electrodeposition of metal nanoparticles are the three methods to decorate nanoparticles on surfaces. The sensing mechanism of Au-doped MoS_2_ is elucidated by Parlak et al. [48]. Au nanoparticle-doped materials proved to have better sensitivity to glucose. Au nanoparticles for glucose sensing enhancement were also studied by Su et al. [49]. The difference between Parlak et al. and Su et al. is that Parlak et al. dispersed Au nanoparticles with ultrasonication while Su et al. synthesized Au nanoparticles directly on MoS_2_ by the hydrothermal method. MoS_2_ nanosheets were dispersed in a 10 mM phosphate buffer solution and were treated with ultrasonication. Both samples were exclusively sensitive to glucose molecules. A vast study is carried out on MoS_2_ substrates by decorating them with different metals, including Ni, Cu, Au, Ag and Pt etc., adopting different synthesis techniques, including liquid exfoliation, solution method, hydrothermal method, and the electrodeposition method has been studied [50,51,52,53]. Bimetallic nanoparticles are difficult to synthesize, and the field is yet to be explored. Similarly, SnS_2_ is an n-type semiconductor having a bandgap of 2.18–2.44 eV and can also be employed for glucose and H_2_O_2_ detection [54]. They are resistant to acids and have good stability in the air, which has the capability of photocatalysts. After fabrication, the reproducibility and stability of the MWCNT–SnS_2_ sensor are demonstrated in Figure 5. Amperometric responses were shown and responded only to glucose while having no response to ascorbic acid and uric acid. All solutions were 0.1 mM in pH 6.0 phosphate buffer solution applying 0.43 V.

### 2.2. Factors Influencing the Electrochemical Glucose Sensing Performance and Monitoring

The electrical conductivity and geometry of the electrode material are the most important factors influencing an enzyme-free electrode’s electrochemical sensing potential. The higher the conductivity, the more electronic transfer and molecular adsorption there will be. The active sites accessible for glucose oxidation are determined by the shape and size of the electrode. Other factors that influence non-enzymatic glucose sensing potential include environmental conditions such as temperature, pressure, and humidity, physiological states that result in pH shifts and poisoning from species other than glucose molecules (interference effect), the electrode’s stability, reversibility, and biocompatibility, the cost of the material, which should be low in order to reduce the overall fabrication cost of the device, and the accuracy of the fabricated device. To evaluate the accuracy of glucose sensor performance, a set of parameters, including sensitivity, the response time, linear range limit of detection (LOD), operation potential, medium pH, selectivity, and so on, is utilized. Another factor influencing the electrocatalytic activity of enzyme-free glucose sensors is their nanomorphologies, which give a huge surface area for electrocatalytic activities.

When considering electrochemical sensing, the electrochemically active surface area is an important parameter to consider. It is also suggested that the increased surface area of two-dimensional (2D) or three-dimensional (3D) nanostructured materials allows for simple surface functionalization, allowing for an effective charge-transfer process. The main limitation of the non-enzymatic glucose sensing activity is its instability with the prolonged use of under-applied working potential. The 2D working electrode can prevent the sensing substrate from agglomerating, deforming, and collapsing [55,56]. Similarly, electrodes should be used with care to achieve precise glucose sensing. Because the pH of human blood ranges between 7.3 and 7.5, glucose-sensing operations should be carried out in physiological settings to optimize various electrochemical characteristics. Glucose levels in diabetic people vary. Even the same patient can experience significant glucose level fluctuations. They can range between 4 and 6 mM. As a result, a reliable glucose sensor should properly detect glucose concentrations across a wide range.

Ginsberg describes four types of errors that occur during blood glucose detection that can significantly impact detector effectiveness. Variations in strip production, strip storage, and ageing can all impair the accuracy of glucose detectors [57]. The efficiency of glucose detection varies from strip to strip when the strip is used as a source of error. Insufficient enzyme coverage can impair glucose meter performance. Excess enzyme on a thick level is utilized in strips to avoid this possibility of inaccuracy, and so slight decreases in the enzyme level or amount do not affect glucose measurements. Sometimes we receive strips with very small individual reaction wells (the sample chambers) (2–3 mm), and a well-size difference of 50 µm can lead to a 3% error. Similarly, the amount of mediator might have a negative impact on the performance of electrochemical glucose sensors. The electrode oxidizes the mediator, which results in the glucose signal. As a result, mediator decrease due to prolonged usage or inactivity can result in incorrect glucose detection. Glucose strips are constantly subjected to biological reactions. As a result, glucose testing strips always have a shelf life. Aside from these conditions, various additional physical factors, such as altitude and temperature (which are less predictable), have an impact on glucose-sensing capabilities. The only remedy for temperature factors is to provide an error warning under extreme conditions. It was discovered that altitude impaired GOx-based meters by 6–15%, whereas non-oxygen-dependent GDH devices had accuracy within 5%. The accuracy of the five strips tested at 8°C varied significantly, regardless of the enzyme base, with 5–7% errors either positively or negatively. Other sources of errors that cannot be adequately addressed are those caused by the patient while erroneously employing glucose detection strips and variations in hematocrit, the ratio of red blood cells to plasma in whole blood. The glucometer’s superior design can avoid the former inaccuracies, while the latter vary greatly in the blood. To avoid contamination and hence ensure accurate glucose testing, the patient should maintain proper cleanliness. The fourth source of mistakes, pharmacological sources of errors, is usually minimal when different drugs can influence the measured glucose level. Other sugars, such as maltose and xylose, can interfere with glucose dehydrogenase. The most hazardous chemical, however, is icodextrin.

### 2.3. Fabrication Procedure

Attaching nanostructures to the surface of the electrode while keeping their designed morphology, large surface area, and physicochemical properties for promising sensing applications is always a huge issue. Various synthesis procedures can be used to insert the catalysts onto a suitable surface to construct 1D, 2D, and 3D electrodes for enzyme-free glucose bio-detection. In 2008, Lee et al. fabricated a non-enzymatic silicon CMOS (complementary metal-oxide-semiconductor) integrable micro-biosensor [58]. Many advantages exist for silicon CMOS integrable microsensors, including reduced cost, smaller size/volume, low power consumption, mass production and implantable devices. First, an insulating layer was deposited over the silicon substrate. Using the dry-etching process, a Ti/Pt film was sputtered over the SiO_2_ layer and patterned to generate three distinct electrodes (counter CE, working WE and reference electrodes RE, Figure 6) of a three-electrode system. After that, the Pt WE were electroplated with nanoporous Pt. After introducing the sample into the C16EO8 mixture, liquid crystal templates of C_16_EO_8_ were generated over the Pt layer by lowering the temperature from 85 °C to various degrees. On top of the flat Pt electrode, a liquid crystalline hexagonal structure was produced at this stage. After removing the surfactant mould in deionised water, the nanoporous Pt electrode was created. Finally, the micro-sensor was created by screen printing Ag/AgCl paste over the RE. By comparing RFs and reproducibility of nanoporous Pt electrodes, the ideal electroplating conditions were established to be 35 mC/mm^2^, −0.12 V, and 25 °C, respectively. Increasing the applied voltage, charges, and temperature would reduce the roughness factor (RF) of Pt-based electrodes, leading to an increase in Pt grain size and overplating of the nanomolds generated. The optimized results reveal that the produced CMOS integrable non-enzymatic micro-sensors are promising for tiny handheld health care systems and continuous monitoring system applications in vitro and in vivo.

The cyclic voltammetry approach can be used to create disposable screen-printed carbon/nickel composites over indium tin oxide (ITO) electrodes for glucose detection without the usage of enzymes (DSPNCE) [59] (Figure 7). The prepared electrodes were notable for their lack of interference from common physiologic interferents such as ascorbic acid (AA) or uric acid (UA). As a result, this approach enabled the fabrication of a simple, disposable glucose biosensor.

The electrochemical deposition was used to successfully prepare cuprous oxide nanoscale thin-film electrodes with various shapes [60]. According to the results, the synthesized cuprous oxide had a greater purity, a tidy morphological structure, and a consistent grain size. This thin-film sensor having a sword-shaped dendrite, responds well to glucose and shows potential for use in sensors. Furthermore, it has a good linear relationship in the glucose concentration range of 1–20 mgL^−1^ with a detection limit of 0.337 mgL^−1^, a sensitivity of around 23.24 mA cm^−2^ mM^−1^, and high stability. For the first time, the template-assisted electrodeposition technique was used to create high-performance Co–Ni–Cu alloy nanotubes for non-enzymatic glucose detection [61]. The electrodeposition voltage and duration were adjusted for the production of high-quality ternary alloy nanotubes. The great performance is due to the high surface area of nanotube arrays as well as the synergistic effects of Ni, Co and Cu metal components. Furthermore, they demonstrated strong dependability and anti-jamming capability. Tafel plots revealed that alloying increased the exchange current density and reaction speed. These synergistic effects can also be inserted into 2D electrodes by alloying for improved glucose detection. Electrochemical techniques are also employed to make Ag/CNTs nanocomposite electrodes for non-enzymatic glucose sensors [62]. The SDS and COOH functionalized CNTs solution on the FTO electrodes generated an extremely dispersed solution for uniform and homogeneous spin coating of CNTs. Silver nanoparticles of different concentrations were electrodeposited on the manufactured CNTs electrode using cyclic voltammetry. The CNTs-coated electrode serves as a support system for the Ag nanoparticles’ highly electroactive surface area. During electrochemical analyses, the identification of glucose molecules improved as the number of Ag particles increased. These nanoparticles could be used to replace pricey electrodes in electrochemical glucose detection. The SDS and COOH functionalized CNTs solution on the FTO electrodes generated an extremely dispersed solution for homogeneous and uniform spin coating of CNTs. Silver nanoparticles of different concentrations were electrodeposited on the manufactured CNTs electrode using cyclic voltammetry. The CNTs-coated electrode serves as a support system for the Ag nanoparticles to enhance the highly electroactive surface area. During electrochemical analyses, the identification of glucose molecules improved as the number of Ag particles increased. These nanoparticles could be used to replace pricey electrodes in electrochemical glucose detection.

Electrodeposition onto a composite electrode consisting of reduced graphene oxide (rGO) mounted on a Ni foam substrate can be used to successfully manufacture uniform and porous CoNi_2_Se_4_ (prepared hydrothermally) [63]. This CoNi_2_Se_4_–rGO@NF hybrid electrode was used as an electrocatalyst for direct glucose oxidation, resulting in a high-performance non-enzymatic glucose sensor. The research also highlights the impact of decreasing anion electronegativity on improving electrocatalytic efficiency by lowering the voltage required for glucose oxidation and increasing the electrode’s selectivity for glucose molecules. A Cu_x_O–NiO nanocomposite film was made using a unique modifying approach for non-enzymatic glucose determination [64]. For 15 min, an anodized Cu electrode was immersed in a solution of H_2_SO_4_, NiSO_4_, and CuSO_4_. Then, in a mixed solution of NiSO_4_ and CuSO_4_, a cathodization process with a step potential of -6 V was commenced, resulting in the creation of a porous Ni–Cu layer over the bare Cu electrode by electrodeposition facilitated by the release of H_2_ bubbles serving as soft templates. The electrodeposition procedure was optimized using experimental design software. The sensor showed high selectivity against some usual interfering species and high stability.

The unique nanostructure and high specific surface area of slack porous nano-TiO_2_ for efficient bio-molecule immobilization attract glucose sensors for electrode manufacturing. As a result, a sensor with an extremely low potential of 0.2 V was constructed using a nano-TiO_2_ coated aluminum tip electrode [65]. Surprisingly, Cu_2_O nanocubes [66] are also developed using the wet chemistry technique. A sensitivity value of 1040 µA/mMcm^2^ was evaluated for the electrode in the linear range from 0.007 to 4.5 mM, and a LOD of 3. 1 µM (S/N = 3). The electrode also showed satisfactory sensitivity towards other molecules, including sugar and ethanol.

## 3. 2D Materials for Electrochemical Glucose Sensing

### 3.1. Graphene

Since Novoselov and Geim discovered graphene in 2004, it has been a prominent carbon material. Graphene is a type of carbon material that features a honeycomb-like crystal lattice and an ultrathin one-atom-thick sheet structure. Graphene-based materials show potential application in electrochemical glucose sensing. Table 3 shows the properties of graphene that facilitate glucose sensing. The quantity of research publications and reviews on graphene and graphene-hybrids for electrochemical sensing has been steadily increasing in recent years. Different research groups have thoroughly examined graphene-based electrochemical glucose sensors. Zhang et al. [18] reviewed the current achievements in the preparation of graphene-based material for electrochemical glucose sensing. G. Gnana Kumar reviewed non-enzymatic electrochemical glucose sensors based on graphene [18]. According to these reviews, graphene is a promising material for electrochemical glucose detection. Comparing GO and rGO, pristine graphene has less interfacial contact; hence GO and rGO were widely utilized in high-performance glucose sensors. Since the application of graphene-based electrochemical glucose sensors was already reviewed in 2018 and 2017, we will focus here on the studies done after 2018.

#### 3.1.1. Enzymatic Glucose Sensors Based on Graphene

Enzyme immobilization on various nanostructured materials (organic or inorganic composites, nanoparticles and 2D materials) has recently received considerable attention in electrochemical sensors because it improves enzyme activity and selectivity towards specific target analytes. Considering the combination of an organic or inorganic composite with a 2D material, Baek et al. modified the gold chip with Cu nanoflower- gold NPs-decorated GO nanofibers. Here GO nanofibers were electrospun over a gold chip that was further decorated with Au nanoparticles following Cu nanoflowers; then, the gold chip was covered with 1% Nafion as a binding agent. With the addition of glucose, it interacts with O_2_ in the presence of Cu-nanoflower@AuNPs-GO NFs to form H_2_O_2_ using an enzymatic catalytic process, resulting in a significant increase in electric current. The doping of AuNPs over GO NFs helps to enhance the sensitivity of glucose. The growth mechanism for Cu-nanoflower follows the nucleation and growth phases. While the Cu^2+^ ion interacts with the phosphate anion in PBS solution to form a copper phosphate crystal, the amide backbone in proteins (HRP and GOx) coordinates with the crystal to form the Cu–protein complex. This complex acts as a seed for the nanoflower, and the nuclei expand over time to form the nanoflower’s petals. A multi-layered nanoflower structure might be formed by a sequence of these successive processes. The Cu-nanoflower reacts with the glucose itself, forming a synergetic action with the GO NFs and AuNPs. Its stability was bolstered by the GO NFs as a backer. Because the 3D-structured AuNPs increase the surface area of the AuNPs, an intrinsic peroxidase-like activity could help HRP improve the electrochemical activities [67]. The immobilization of the enzyme over the nanoflowers such as Cu helps to enhance the sensitivity, durability as well as stability. In addition to this, GO NFs as a supporter also enhance the stability. This sensor shows excellent current even after 20 days suggesting the potential for field use.

Another electrochemical sensor was fabricated by Mao et al. He used rGO to improve the sensitivity and selectivity of the ZnO nanorod biosensor. In this work, the ZnO nanorods were hydrothermally synthesized over a polyethylene terephthalate (PET) substrate. The ZnO/PET working electrode was then coated with electrodeposited rGO, and AuNPs were scattered on the surface, resulting in ZnO/rGO/Au/PET. Lastly, the GOx was physically adsorbed on the electrode’s surface, yielding a GOx/rGO/ZnO/Au/PET glucose sensor with a sensitivity of 56.32 µA mM ^−1^ cm^−2^ and a linear range of 0.1 to 12 mM [68]. However, under 10 cycles of bending, this working electrode exhibits good current responsiveness. However, after 15 cycles, the device’s performance is poor because GOx and rGO peel away from the ZNO nanorod, resulting in microcrack development [68]. Hossain and Slaughter have suggested a hybrid glucose biosensor employing MWCNTs and graphene with great sensitivity and selectivity. A one-step solvothermal approach was used to make a solution containing both MWCNTs functionalized with carboxylic groups and chemically-derived graphene. PtNPs were electrochemically deposited onto a thin layer formed by drop-casting this suspension onto an Au electrode. Finally, GOx was immobilized and coated with Nf on the nanostructured electrode. The hybrid biosensor was built with a sensitivity of 26.5 µA mM^−1^cm^−2^ and a linear detection range of 0.5 to 13.5 mM [69].

#### 3.1.2. Non-Enzymatic Glucose Sensors Based on Graphene

A large number of transition metal nanoparticles such as platinum (Pt), gold (Au) and palladium (Pd) were synthesized and employed in non-enzymatic sensors. These metal nanoparticles have an electronic structure having unpaired-d electrons, and unfilled d-orbitals help in the electrocatalytic activity of glucose. In order to enhance the electrocatalytic activity of glucose, several supports have been investigated, among which GO serves as a suitable supporting material. Because of its high catalytic activity and stability, platinum is commonly employed as an electrocatalytic electrode material. Sakar and colleagues describe a unique design of glucose sensor with graphene Schottky diodes consisting of a graphene (G)/platinum oxide (PtO)/n-silicon (Si) heterostructure, taking into mind the benefits of combining graphene with the platinum electrode. They discovered that the platinum oxide film thickness affects the sensor’s sensitivity and that raising the PtO film thickness can increase the sensor’s sensitivity by up to 150%. This was ascribed to an increase in the number of active sites for glucose oxidation and the thickness of the graphene layer, resulting in increased charge carrier mobility and concentration. The device works by glucose molecule oxidation over the surface of the suggested heterostructure electrode due to the catalytic activity of the PtO thin film, as illustrated in Figure 8. Aside from that, the glucose molecules are broken down into gluconolactone, which creates gluconic acid, H_2_, and electrons. The current was measured from the graphene surface after applying a forward bias to the Si terminal. The physisorption interaction between PtO (metal-semiconductor Schottky junction) and graphene caused a shift in graphene’s fermi-level location and p-doping [70]. Here graphene is employed as both a protective and sieving layer to protect the PtO film and further enhance the sensor stability and selectivity.

Compared to Pt-based electrodes, the main advantage of using Au-based electrodes for glucose sensing is the higher current response, which allows for higher sensitivity and the ability to detect glucose at a neutral pH. However, the main drawback of Au-based electrodes is the low glucose oxidation efficiency on the Au electrode surface, which can be mitigated by using arrays of nanoelectrodes separated by non-electroactive materials. Furthermore, because these electrodes are better activated in alkaline solutions, they cannot be used in in vivo studies, they have surface contamination from anions such as phosphates and chlorides, and their selectivity is significantly smaller than Pt-based electrodes. Scandurra et al. prepared a graphene paper-based electrode using the dewetting technique. In this case, an 8-nm-thick Au layer was sputtered onto graphene paper before being dewetted with a laser. Dewetting with a laser-produced smaller AuNPs on the electrode surface. The sensor’s sensitivity was 1240 µA mM^−1^ cm^−2^ [71].

Combining the advantages of another transition-metal oxide palladium (Pd) and CNT with graphene nanoplates, Kiattisak et al. modified glassy carbon (GCE) using a nanocomposite of multi-walled carbon nanotubes wrapped with palladium nanoparticle-graphene nanoplatelets (PdNPs-GNPs/MWCNTs). This modified GCE was coupled with a flow injection amperometric detector where this sensor exhibits a detection limit of around 0.008 mM. The following reactions can be used to describe the mechanism of glucose oxidation by Pd nanoparticles.
Pd + Glucose → Pd-Glucose _ads_ + H^+^ + e^−^(1)
Pd (OH)_2_ + glucose → Pd + gluconolactone + H_2_O(2)
Pd + 2OH^−^ → Pd (OH)_2_ + 2e^−^(3)

Electrode materials such as Au and Pt are suitable for glucose detection, but they are costly. As a result, non-precious transition metals such as nickel (Ni) and copper (Cu) and their oxides have been studied. Despite the wide range of catalytic materials available for glucose sensors, nickel (Ni)-based nanomaterials have attracted researchers’ interest due to their high catalytic activity for glucose oxidation in alkaline medium, resulting in NiOOH and Ni(OH)_2_ species. Despite their high electrocatalytic activity, surface fouling from glucose oxidation compromises their stability. As a result, a conductive nanostructured substrate is required to improve electron transport and electrode stability against Ni glucose oxidation. Considering graphene’s excellent properties as a catalyst support material, Jothi et al. fabricated a non-enzymatic glucose sensor based on graphene nanoribbon/graphene sheet/nickel nanoparticles to prevent re-stacking and increase surface area. This Ni-based hybrid has a larger specific surface area, more active sites, and better electrical conductivity, allowing for easy ion transport and unrestricted OH- diffusion throughout the electrochemical process. The mechanism of glucose oxidation by Ni nanoparticles can be explained using the reactions listed below [72].
NiO(OH) + glucose → Ni(OH)_2_ + gluconolactone(4)
Gluconolactone → gluconic acid(5)

Among these Ni-based materials, Ni-based porous materials have a large surface area, and more active sites have been used as sensitive materials for glucose sensors. To exhibit the full potential of porous Ni materials, different kinds of substrates such as Ni foil, SPEC, and Cu foil have been used. Still, the electron transferability of porous Ni is still weak. Exploiting the properties of graphene, where graphene provides anchoring, conducting and separating actions, Ren et al. prepared a porous Ni over exfoliated graphene by using the hydrogen bubble method over the Cu foil substrate. This porous sensor shows high sensitivity due to the high electron transfer ability between the active reaction centre and electrode induced by the graphene [73]. Another Ni/graphene hybrid material was reported by Lavanya et al. where this hybrid material was synthesized using a simple in situ chemical reduction method. To begin, Ni^2+^ was adsorbed onto the GOR and GOS surfaces via the electrostatic interaction between functional groups (epoxy, carboxyl, and hydroxyl groups) and the Ni^2+^ on the GOR and GOS surfaces. The Ni(OH)_2_ developed in GOS after adding NaOH precipitant was then co-reduced with the help of hydrazine hydrate. The reducing agent contributes to the reduction of GOR and GOS, the conversion of Ni^2+^ ions to Ni^0^ nanoparticles, and the development of Ni nanoparticles in the GS/GNR network throughout this process. This sensor shows a detection limit of around 2.5 nM, and this superior catalytic performance of this sensor was due to the synergistic action of Ni nanoparticle and GS/GNR hybrid network. The Ni nanoparticles were uniformly dispersed throughout the GS/GNR hybrid to increase electrical conductivity, a greater surface area with more active sites, and unrestricted flow of OH- ions during the electrochemical process. The incorporated Ni nanoparticles are an effective catalytic active material for direct glucose oxidation, which improves electron transfer and results in a high lower detection limit, selectivity, and good sensitivity for glucose detection. Here during the co-reduction of GOR, GOS and Ni(OH)_2_, the abundant functional groups in the basal and edges of GOR and GOS offer additional anchoring sites for the incorporation of more Ni nanoparticles, i.e., catalytic site, and increase the rate of electron transfer between the electrode and glucose and thereby enhancing the performance of sensor [72].

According to recent research, bimetallic materials, particularly bimetal alloys, outperform monometallic counterparts in catalytic performance. Several studies on the performance of bimetallic materials for glucose sensing have been published. However, its performance can be improved by using a support material such as graphene. Deng et al. created a NiFe alloy nanoparticle/graphene oxide hybrid (NiFe/GO) based on this [74]. Rukiye et al. described another Ni-based bimetallic material in which the working electrode was decorated with monodisperse platinum–nickel nanocomposites-decorated on reduced graphene oxide (Pt/Ni@ rGO) manufactured utilizing a novel ultrasonic hydroxide aided reduction technique. This sensor shows excellent electrochemical activity with a sensitivity of around 171.92 μA mM^−1^ cm^2^ and LOD of 6.3 μM [75].

Among the transition metallic nanoparticles, metal sulfides, which are found as minerals in nature, have higher cycle stability than conducting polymers. Furthermore, because of the numerous sorts of structures, they have a lot of research potential in glucose sensors. Among these, CuS has attracted a lot of attention due to its exceptional fundamental property diversity, such as attractive photovoltaic capabilities, strong thermal stability, great transport properties, and so on. Yan et al. prepared CuS nanoflakes reduced graphene oxide (rGO/CuSNFs) nanocomposite to avoid its coagulation using a surfactant-free method. This sensor shows excellent sensitivity and a low-detection limit [76]. On the other hand, aggregation and low dispersion remain major issues in the synthesis of MNPs on graphene support, resulting in poor electrocatalytic performance and low stability.

Intrinsically conducting polymers (CPs) are one of the most relevant and extensively used materials for sensor modification due to their unique chemical and physical properties, such as adjustable architecture, adaptability, versatility, room stability, and sensitivity to surface changes in electrochemical activity with minor changes in its surface. Polypyrroles, polyanilines, and polythiophenes have received a lot of attention because of their good film-forming properties, electrical semiconductivity, great transparency in the visible range, and exceptional thermal and environmental stability [77]. CPs have recently been hybridized or mixed with graphene-based materials. When the materials are linked, each component’s combined optical, electrical, thermal, mechanical, chemical, or electrochemical capabilities can be exploited for chemical and biological sensing [78]. For glucose measurements, a composite comprising (PANI/PDPA) along with graphene nanosheets synthesized in a liquid–liquid (CHCl_3_/HCl) interface individual by Muthushankar et al. During the polymerization procedure, the PANI and PDPA chains grew nicely and were evenly distributed over the graphene nanosheets. The improved electro-catalytic activity of Gra-PANI-co-PDPA-ME towards glucose was achieved with greater sensitivity of 0.51 µA/µM at 5 s. The presence of aniline, graphene and diphenoquinone diamine (DPDI^2+^) together with their synergistic relationship improves electron transport for glucose oxidation. Interference tests further confirm that the constructed sensor is best suited for glucose sensing performance [79]. Silver (Ag) nanoparticles are considered a promising candidate because of their excellent stability and biocompatibility. Utilizing the advantages of this, Ag nanoparticle and carbon-based nanocomposite having excellent electrical conductivity and stability, Deshmukh et al. demonstrated an Ag-PANI/rGO nanocomposite using a simple hydrothermal method. This composite demonstrated a detection limit of 0.79 µM with a quick response time. They proved the viability of using Ag-PANI/rGO nanocomposites to detect glucose in real-world samples of organic fluids (milk, apple juice, mango juice, orange juice, and Coke). Here the introduction of rGO into PANI polymer improves the operational stability and electron transfer rate. This sensor shows stability up to 30 days [80]. Poly(3,4-ethylenedioxythiophene) (PEDOT) is another candidate that has piqued the interest of this group of CPs. PEDOT has been extensively used as an electrode material among CPs due to its low oxidation potential, good stability and mid-band gap energy. Considering the favorable properties of this polymer, a non-enzymatic glucose sensor based on an Au electrode was modified utilizing electroreduced graphene oxide (ERGO) and layered PEDOT by Mesut et al. The thin films of PEDOT–ERGO were made using a simple electrochemical approach based on progressively layering electrodeposition on an Au electrode, which was employed as a non-enzymatic glucose sensor in this study. The PEDOT–ERGO nanocomposite modified Au electrodes (Au–PEDOT–ERGO) were employed as electrocatalysts for voltammetry and amperometry glucose detection. This sensor exhibits a LOD of around 0.12 μM with a sensitivity of 696.9 μA mM^−1^ cm^−2^ [81].

### 3.2. MXene

MXene is suitable for building high-performance electrochemical glucose sensors because it has high hydrophilicity due to surface termination groups (O, OH, and F), great electrical conductivities, excellent ion intercalation behavior, facile functionalization, and dependable large-scale manufacture. Several high-quality reviews on MXenes have been published to date, suggesting the applicability of MXene in sensor fabrication [82,83,84,85]. However, no review focused on MXene-based electrochemical glucose sensors is available so far. Table 3 shows the properties of MXene facilitate in glucose sensing

#### 3.2.1. Enzymatic Glucose Sensors Based on MXene

MXenes, similar to other 2D materials, can include additional materials such as metal nanoparticles, enzymes, CPs, and metal oxides, resulting in improved structural and electrical characteristics [86]. For example, without the need for external reducing agents, one-step hybridization of Au, Ag, and Pd nanoparticles from their respective aqueous solutions onto the surface of MXene was performed and employed as a substrate for SERS. Ti_3_C_2_ was the first MXene to be employed in constructing electrochemical glucose sensors, and Rakhi and her colleagues produced the first electrochemical sensor based on Ti_3_C_2_ in 2016. A nanocomposite of gold nanoparticles (Au) and MXene (Ti_3_C_2_T_x_) for glucose detection has been reported. After in situ reductions of chloroauric acid with sodium borohydride, they deposited Au nanoparticles on the surface of MXene, and the nanocomposite was subsequently dispersed in Nafion. This composite was then drop casted on a glassy carbon electrode before being immobilized with glucose oxidase (GOx) to produce Au/GOx//MXene/Nafion/GCE. Au nanoparticles were critical in increasing the electron exchange between the electrode and the active center of GOx. After a first morphological examination, the biosensor was assessed using CV and amperometry. The glucose-sensing mechanism is based on the following Equations (6) and (7). GOx is composed of two extremely similar protein subunits and one coenzyme molecule, flavin adenine dinucleotide (FAD). FAD is reduced to FADH_2_ during the transfer process with 2e^−^ and 2H^+^ because it is present in the active site of the GOx. The biochemical reaction between GOx and glucose culminates in the conversion of glucose to glucono-D-lactone and the reduction of FAD to FADH_2_. FADH_2_ was then oxidized by dissolved oxygen to create H_2_O_2_ and form FAD.
GOx (FAD) + Glucose → GOx (FADH_2_) + glucono-D-lactone(6)
GOx (FADH_2_) + O_2_ → GOx (FAD)+ H_2_O_2_(7)

MXene, as well as Au nanoparticles, can both contribute to improved electron transfer kinetics between the electrode and active redox centers of the enzyme, while the large surface area with the distinctive layered architecture of Au/MXene nanocomposite effectively accommodates the enzyme. The amperometric response was shown to be linear over a concentration range of 0.1 to 18 μM, with a LOD as low as 5.9 μM. This sensor also shows long-term stability. Furthermore, in the presence of AA, UA and DA, this suggested biosensor is highly specific for glucose. The use of enzymes for glucose detection ensures a strong electrochemical response, particularly in complicated matrices. However, there are significant drawbacks to using enzymes, including the risk of enzyme inactivation, the reproducibility of enzyme immobilization, and the increased cost of analysis. Hence, the development of non-enzymatic sensors is becoming increasingly popular [87]. Using a mixing–drying process, researchers improved biosensing capabilities by combining MXene nanosheets with hydrophilic groups with the properties of graphene sheets to offer new functions. This three-dimensional (3D) porous hybrid film had a more open structure, which allowed glucose oxidase to enter through the inner pores, improving the hybrid film’s stable immobilization and retention of GOx. The constructed biosensor outperformed conventional 3D porous materials with sensitivities of 20.16 mM (LOD 0.13 mM) and 12.10 mM (LOD) 0.10 mM) in O_2_^−^ saturated phosphate-buffered solution (PBS) and air-saturated PBS, respectively. This 3D porous film results in higher stability for GO_X_ immobilization. This is due to (i) the enhanced hydrophilic property of this hybrid MG film, (ii) the open surface facilitates more access to GO_x_, and (iii) the excess 3D pores and hydrophilic wall of film this film provides a favorable microenvironment for GO_X_ to stay in the pores and avoid re-dissolution in PBS [88].

In practical applications of glucose biosensing, the harmful intermediate product, H_2_O_2_, formed during enzymatic glucose oxidation usually hinders the action of GOx. Wu, M. et al. created a hybrid Ti_3_C_2_/poly-L-lysine (PLL)/glucose oxidase (GOx) nanohybrid that could catalyze the cascade processes of glucose oxidation and the intermediate H_2_O_2_ breakdown to address this issue. The PLL-modified MXene possesses a positive charge and exhibits excellent GOx loading capacity where the amine-groups in PLL are able to form a crosslink with the physically adsorbed enzyme to form MXene nanosheets covered with GOx_/_PLL. The Ti_3_C_2_ MXene was shown to be capable of catalyzing the breakdown reaction of H_2_O_2_, and when combined with the GOx, a cascade reaction in glucose decomposition was created. Ti_3_C_2_/PLL/GOx nanoreactors with higher catalytic activity were placed on glassy carbon electrodes to create a 2.6 μM LOD glucose biosensor [89].

#### 3.2.2. Non-Enzymatic Glucose Sensors Based on MXene

Li et al., for the first time, reported a non-enzymatic glucose sensor based on MXene/Nickel–Cobalt layered double hydroxide (NiCo-LDH). They synthesized MXene over NiCo-LDH using a simple hydrothermal method. Because MXene is negatively charged, it can absorb cations and provide nucleation sites to form nanoparticles. The redox reaction in the following equation produces hydroxyl ions, which react with Co^2+^ and Ni^2+^ to make hydroxide monomers on the MXene sheet, and the monomers then react with others to produce primary particles in the hydrothermal reaction. In the meantime, some divalent cobalt is oxidized to trivalent cobalt. As the number of initial particles increased, they progressively collected and transformed into nanosheets coated on MXene.
4CH_3_OH + NO_3_^−^ → 4HCHO + NH_3_ + OH^−^ + 2H_2_O(8)

This glucose sensor has a broad linearity range (0.002 mM–4.096 mM), a LOD of around 0.53 µM, with a quick response time (3 s) at a working potential of 0.45 V (vs. SCE). Additionally, good selectivity, stability, and repeatability were obtained [90]. A non-enzymatic glucose sensor based on the MXene–Cu_2_O hybrid was recently studied by Gopal et al. This sensor has a linear range of 0.01–30 mM and an LOD of 2.83 µM [91].

### 3.3. TMD Based Electrochemical Glucose Sensors

Two-dimensional TMD nanosheets are appealing as electrochemical glucose sensors because of their high conductivity, large surface area, high signal/noise ratio and quick electron transfer kinetics and, most importantly, their practicality for forming composites.

#### 3.3.1. Enzymatic Glucose Sensors Based on TMD

Other metal NPs, such as Au NPs, are utilized to alter MoS_2_ in addition to Ni and Cu NPs in enzymatic glucose sensors [92]. The use of AuNPs accelerates electron transport from the electrode to the immobilized enzyme. This enables GOx electrochemistry without the usage of an electron mediator. Thus, the GC modified with Au NPs and MoS_2_ was studied by Su et al. and showed enhanced electrocatalytic activity [93]. Another electrochemical glucose sensor was fabricated by Parlak et al. using a MoS_2_/Au NPs hybrid. Here, enzymes aided in directly transferring ions to the modified MoS_2_ surface, increasing its bioelectroactivity for redox processes. Figure 5 depicts a typical biosensor fabrication procedure in which AuNPs were self-assembled over the MoS_2_ nanosheets and subsequently connected to GOx for glucose detection in the presence of a mediating agent, ferrocene carboxylic acid. Because of their open 3D architectures with a significant interlayer space and surface area for enzyme immobilization, these electrodes demonstrated remarkable sensitivity to glucose. As a result, fast mass transport was accelerated, and glucose diffused quickly across the electrode surface. Despite the fact that the study revealed bioelectrocatalytic reactions of glucose, suggesting substantial improvements in biocatalysis at a MoS_2_-based enzymatic nanointerface, real-sample analysis and interferences were not performed, which could have demonstrated selectivity and further justified the analytical functionality of the biosensor. The constructed MoS_2_ enzymatic electrode, on the other hand, revealed a new potential for creating a unique 3D sensing substrate [48].

Because H_2_O_2_ is a consequence of many oxidative biological activities, measuring it with MoS_2_ electrodes allows for the identification of many more tiny molecules by integrating the sensor into lab-on-chip devices for intracellular detection. For instance, a glucose sensor based on MoS_2_ demonstrated strong activity towards H_2_O_2_. GOx was immobilized over the surface and used to detect glucose. The current response of this biosensor grew as the concentration of glucose climbed from 2.0 to 16.0 mM. Despite the fact that glucose oxidation consumes O_2_, the biosensor’s overall current response increased because both O_2_ and H_2_O_2_ contributed to the increase in reduction current. To explore the large surface area and enhanced electrical conductivity of 3D porous graphene aerogel, Jeong et al. fabricated a flow-injection biosensor device. They incorporated 2D MoS_2_ with 3D graphene aerogel using a facile hydrothermal approach to enhance the electrochemical sensing performance. Here the different porous structures of 3D MGA provide rapid, efficient pathways for electrons and ions, demonstrating excellent electrochemical performances. The following qualities contribute to the superior performance of these 3D MGA-based biosensors. At the interface, the interconnected network of MoS_2_ exposes extensive basal planes against the electrolyte solution, which enhances the H_2_O_2_ reduction activity of MoS_2_ nanosheets using oxygen substitution at exposed Mo edges. Thus, showing excellent selectivity and sensitivity [94,95].

#### 3.3.2. Non-Enzymatic Glucose Sensors Based on TMD

Many researchers have worked ceaselessly on sensor fabrication for glucose detection employing MoS_2_ nanosurfaces among TMDs [96]. Massive efforts have been undertaken to tackle MoS_2_ difficulties such as low electrical conductivity and limited electrical transportation. The extended technique for other heteroatom doping such as B and N for increased semiconducting characteristics and catalytic reaction revealed an intriguing advancement with doping of Ni in the MoS_2_ framework. For the first time, Huang et al. fabricated an electrochemical glucose sensor based on Ni–MoS_2_ hybrid [50]. As an extension of this work, a functional hybrid of Ni-doped MoS_2_ and rGO was later fabricated. The electrochemical analysis demonstrated that the Ni-doped MoS_2_/rGO composite has a high electroactivity due to the efficient electron transport rates, highly exposed catalytic sites, electrical conduction efficiency, and large effective surface area of MoS_2_ and rGO. After successfully detecting glucose in human serum samples, the sensor demonstrated its great analytical value as a prospective catalyst for non-enzymatic investigation of additional metabolites such as cholesterol, H_2_O_2_ and lactose, among others [97]. In another case, Ji et al. modified glassy carbon with nickel (II) hydroxide nanoparticles and MoS_2_ film using an electrodeposition method. Here MoS_2_ act as catalytic support for the nickel (II) hydroxide nanoparticles and thereby enhancing the electrochemical activity of bare MoS_2_ and nickel (II) hydroxide NPs [98]. Huang et al. decorated MoS_2_ nanosheets using Cu nanoparticles. The prepared hybrid using the chemical reduction method and cyclic voltammetric and amperometric studies in alkaline media showed good sensitivity, 1055 µA mM^−1^ cm^−2^ [95]. Cu_2_O electrocatalytic activity has sparked interest in non-enzymatic glucose sensors, similar to Cu electrocatalytic activity. As a result, a new combination of 3D MoS_2_ nanoflowers and Cu_2_O metal oxide was developed for non-enzymatic glucose detection using amperometry. The electrode’s exceptional electrochemical activity could be attributed to synergistic effects offered by the peculiar structural characteristics of Cu_2_O–MoS_2_ nanohybrid enveloped in 3D flower-like MoS_2_ with thin interconnecting nanosheets and largely scattered Cu_2_O. The huge surface area of MoS_2_ nanosheets allowed for substantial Cu_2_O dispersion, boosting the amount of electrochemically active sites and, as a result, electrical conductivity and fast heterogeneous electron transfer rates. This sensor shows excellent sensitivity with an LOD around 1 µM [99]. Wu and their group have demonstrated the first MoS_2_-nanosheet-based electrochemical sensor. In the redox systems, exfoliated MoS_2_ nanosheets were electrochemically reduced in NaCl solution to generate reduced MoS_2_ (rMoS_2_) with a fast electron transfer rate and strong conductivity. The generated rMoS_2_ was used to detect glucose by immobilizing GOx and to specifically detect dopamine (DA) in the presence of UA and AA, as shown in Figure 9A. The reduction current at 0.33 V, arising from oxygen reduction, decreased as the glucose concentration increased from 0 to 20 mM [100].

Several reports on electrochemical glucose sensors based on MoS_2_ are available. Other TMD-based glucose sensors, on the other hand, are still being investigated. For the first time, Gayathri et al. reported a MoSe_2_ based non-enzymatic glucose sensor. Using a simple hydrothermal method, they decorated these MoSe_2_ nanosheets over a NiO nanorod. This composite modified GCE shows excellent electrochemical activity. The electro-catalytic mechanism for glucose oxidation is as given in following Figure 9B, and it follows according to the equation given below:NiO + OH^−^ → NiO(OH) + e^−^(9)
NiO(OH) + glucose → Ni(OH)_2_ + gluconolactone(10)

This composite had a linear response for glucose detection from 50 mM to 15.5 mM, with a LOD of 0.6 µM [101]. Another transition metal selenide-based non-enzymatic glucose sensor was reported by Mani et al. for the first time. The synthesized NiSe_2_ nanosheets using a facile-hydrothermal approach and modified the GC with these nanosheets show excellent activity towards glucose [102]. Comparing transition metal sulfides and selenides, tellurides possess lesser electron negativity, large atomic size, high conductivity, and less toxicity. Due to these peculiarities, transition-metal tellurides received considerable attention in the field of biosensors. For example, Ni_3_Te_2_ nanostructure was reported as a successful composite for non-enzymatic glucose sensing [103]. Similarly, CoTe_2_ nanosheets, which were grown over a 3D nickel foam was, also show excellent electrochemical activity and used for glucose sensing. Among the transition metal tellurides, CoTe_2_ exhibits exceptional electrochemical performance and high stability, making it an excellent candidate for glucose sensing fabrication [104].

### 3.4. Layered Double Hydroxides (LDHs)

Nanostructures, having a large surface area and a rough surface, may be capable of transmitting faraday currents for oxidation while reducing the impact of external interfering molecules, such as UA, AA, and AP. LDHs are a type of two-dimensional material with the structural formula [M_1−x_ ^II^ M x^III^(OH)_2_](A^n−^)_x/n_.mH_2_O, where M^II^ and M^III^ are divalent and trivalent metals, respectively, and A^n−^ signifies the anions between the interlayer spaces. The properties required for glucose sensing are shown in Table 3.

#### 3.4.1. Enzymatic Glucose Sensors Based on LDH

Enzyme-based biohybrid materials, which are made by immobilising various oxidoreductases in the LDH host structure, have been used to detect target analytes such as hydrogen peroxide, phenol derivatives, glucose and other inhibitors or enzyme substrates utilizing amperometric transduction methods, primarily chronoamperometry. There have been very few publications on LDH-based enzymatic glucose sensors. Because of its stability, glucose oxidase (GOx) is frequently used as a model to examine various biosensor setups. The major findings from investigations using glucose biosensors based on the immobilization of GOx in LDH matrices are presented here. Physical adsorption over the electrode surfaces produced by gradual evaporation of colloidal suspensions of mixes of LDH and GOx has been the most extensively utilized method for GOx immobilization. The entrapped biomolecules are cross-linked with the help of glutaraldehyde in the vicinity of bovine serum albumin to prevent the enzyme from being released. Glutaraldehyde was not used since composite materials based on chitosan or alginate with Zn/Al-Cl LDH were used instead. This method of deposition allows a substantial and known quantity of enzyme to be entrapped in the clay layer, assuring great response sensitivity. In this regard, Colambari et al. synthesized Mg/Al LDHs containing ferrocene sulfonate or ferrocene carboxylate (Fc-COOH) as interlayer anions using the coprecipitation method, and they built glucose sensors based on the immobilization of GOx in a glutaraldehyde and BSA network. MnO_2_ nanoparticles were effectively integrated into the outer protective glutaraldehyde/BSA membrane to improve the selectivity of this GOx/Fc-COOH LDH/GC biosensor, allowing oxidizable interferents (e.g., acetaminophen, urate, and ascorbic acid) present in the sample solution to be analyzed to be peroxidised before reaching the electrode surface [105]. Shan and colleagues created a biosensor that is extremely sensitive to glucose utilizing Zn/Cr LDH. Ferricinium derivatives reoxidised _FADH2_ to FAD, which is followed by reoxidation of ferrocene (Fc) to Fc^+^ directly at the electrode, enabling glucose measurement under anaerobic circumstances in this biosensor [106]. The biosensor design reported by Shan et al. is based on a “ping-pong” mechanism for transmitting electrons from glucose to the mediator [107]. Mousty et al. developed another glucose sensor based on redox-active Zn/Cr LDH intercalated in the interlayer domain and contained (3-sulfopropyl) ferrocene carboxylate. Chronoamperometry at 0.5 V was used to examine the biosensor’s performance for glucose measurement under anaerobic conditions, and the sensitivity was 65 µA mM^−1^ cm^−2^ in the concentration range of 10–25 µM [108]. GOx was immobilized at the surface of the electrode by entrapment in inert LDH matrices (i.e., NiAl-NO_3_, ZnAl-Cl,) and ZnAl–Cl/biopolymers (chitosan or alginate) [109] composites with the goal of generating glucose amperometric electrochemical sensors. The majority of these LDH/GOx biosensors were made by solvent casting a predetermined amount of LDH and GOx combination onto a glassy carbon electrode or Pt surface. In these investigations, GOx was immobilized on LDH particles through adsorption, and then glutaraldehyde cross-linking was used to inhibit enzyme leaching. Tonelli’s group demonstrated yet another way of GOx immobilization by trapping GOx during the electrogeneration of NiAl LDH films over the Pt electrodes [110]. Farhat et al. recently published an amperometric biosensor based on Co_3_Mn–CO_3_ layered double hydroxide/GOx with biopolymer carrageenan. They used a coprecipitation approach to make this Co_3_Mn–CO_3_ LDH and then impregnated this LDH composite over a carbon felt to make this 3D porous sensor [110]. Zhang et al. revealed the first-ever direct electron transfer of GOx immobilized in Mg/Al hydrotalcite nanosheets, creating new avenues for the development of third-generation GOx glucose sensors employing LDHs as the immobilising matrix [111].

#### 3.4.2. Non-Enzymatic Glucose Sensors Based on LDH

Through coprecipitation and electrodeposition, nickel and cobalt inculcated LDHs were produced and used for non-enzymatic glucose sensing [112]. Using the electro-oxidation processes of Ni ions, the LDH nanostructures outperformed GOD-based biosensors for glucose sensing even in the absence of GOD catalase. Furthermore, simple LDH materials were fabricated, such as porous structured NiFe LDH nanolayers grown over the nickel foam by the simple hydrothermal method and used for glucose detection [113]. Similarly, Zhao et al. fabricated a non-enzymatic glucose sensor using a multi-component sensing system where the electrode is made up of NiCo-LDH and cobalt copper carbonate hydroxide (CCCH) nanorods over a Cu foam. Here the large surface area and excellent conductivity of the Cu foam help in enhancing the electrocatalytic properties. This multicomponent hierarchical nanostructure features a large accessible surface area, as well as effective ion diffusion and electron transfer, thus enhancing the sensing properties of NiCo-LDH [114].

Hai et al. fabricated a NiAl LDH, loaded carbon microcylinder (CMC) composite using microelectromechanical systems (C-MEMS). When combined with electrocatalytic Ni centres, the suggested method demonstrated good performance due to improved conductivity [115]. Furthermore, several LDH nanoarchitectures intercalated with valuable noble metals, such as Au, NPs incorporating NiAl LDH SWCNTs–rGO hybrids exhibited stronger electrocatalytic capabilities for glucose oxidation than others. Good conductivity, generated from electron tunnelling junctions and 3D CNTs/rGO networks, is thought to be the source of these features [116].

When LDH-based catalysts are integrated with conductive carbon supports such as conductive polymers, CNTs, and graphene, the extraordinary properties of materials based on carbon, such as greater active sites, better consistency, and larger surface to volume ratio, can help improve the catalytic activity and stability of LDH layers. Inserting graphene between the LDH interlayers improves conductivity and glucose detection with high sensitivity while also boosting resistance to poisoning in chloride ion solutions. NiCo LDH-rGO [117] nanoribbons and NiFe LDH-rGO nanohybrids are used in this context [118] and were synthesized using electrochemical coprecipitation and deposition, respectively, and their catalytic properties were investigated. The electrodes with the nanostructures had a significant diffusion coefficient and excellent electrochemical catalytic activity towards glucose. Shahrokhian and group developed a core–shell structure to improve the electrocatalytic properties of CoNi-LDHs, notably in terms of shape, by growing CoNi-LDHs NS over a nanoporous tubular Cu(OH)_2_ utilizing a green, simple, and extremely controllable direct three-step in situ process over a GC. The step involves (1) copper film electrodeposition on GCE, (2) copper conversion to Cu(OH)_2_ nanotubes, and (3) CoNi-LDH electrodeposition over Cu(OH)_2_NTs/GCE. The rate-determining steps of the entire electrocatalytic reaction are represented by the equation below, which involves the diffusion of glucose molecules to the metal sites in the core–shell structure, where the OH^−^ travels from the alkaline solution and then permeates into the CoNi-LDH interlayers.
LDH (OH^−^)− M_III_ + Glucose → LDH CO_III_ + Gluconolactone(11)
2CuOOH + 2e^−^ + Glucose → 2CuO + 2OH^−^ + Gluconolactone(12)

This highly porous core-shell nanostructure adheres strongly to the conductive substrate, allowing for electrical and physical contact between the GCE surface and the active components, improving the overall stability and conductivity of the modified electrode. The commercialization capability of this sensor was also evaluated by growing this 3D hierarchical structure over a screen-printed electrode [119].

### 3.5. Other 2D Materials

Balasubramanian et al. synthesized *a* cobalt vacancy-rich Co(OH)_2_ ultrathin nanosheet over a screen-printed electrode and demonstrated its catalytic activity. A practical, simple, and in situ technique for obtaining cobalt hydroxide nanosheets with numerous cobalt vacancies is described here. The cobalt defects significantly enhance the charge transfer rates and increase the number of electroactive sites, resulting in outstanding glucose and L-cysteine oxidation performance. This V_Co_-Co(OH)_2_ electrode has a low detection limit of roughly 295 nM over a dynamic range of 0.4 M–8.23 mM [120]. Sahoo et al. studied the catalytic activity of non-faceted and faceted crystal cupric oxide (CuO) nanoribbons using a microwave and hydrothermal method. Here the non-enzymatic glucose sensor made-up of both non-faceted and faceted CuO crystals shows enhancement in amperometric oxidation current that is proportional to the glucose concentration. Their study shows that the glucose sensitivity of faceted CuO is higher than that of non-faceted crystal CuO due to the 2D thin and polygonal shape of this facet plane CuO. Here the facet plane of CuO, such as (100) and (110) planes, play a prominent role in improving the catalytic activity due to their higher surface energy. The faceted ends give more surface area and more electroactive species and are advantageous for electrolyte ion transferring and exchanging activities. The catalytic activity with glucose is as follows:2CuO(OH)) + glucose → 2CuO + gluconolactone + H_2_O(13)
Gluconolactone → Gluconic acid(14)

The activation of these CuO nanostructures causes the amperometric current to increase with increasing glucose concentration. Here the hopping free-charges in the facet plane of CuO show a significantly lower detection limit of around 58 µM [121].

Other elements of group VA (also known as “pnictogens” or the “nitrogen” group) can likewise have a layered structure. These elements are similar to graphene: phosphorene, bismuthene, antimonene, and arsenene. Ling chia and group, for the first time, explored the use of pnictogens for non-enzymatic glucose detection. Because pnictogen nanosheets have strong carrier mobilities, they may aid in promoting fast electron transfer kinetics for glucose oxidation. In addition, pnictogens have excess lone-paired electrons on the surface and interact with glucose at high pH with good adsorption capabilities; thus, it could also aid in facilitating the adsorption mechanism by bringing glucose molecules into close vicinity with the electrocatalytic Au@Ag nanorods and enhance the sensing capabilities of both pnictogens and Au@AgNR. This study theoretically proves and opens up new vistas for the potential use of pnictogens-based composites in the future development of electrochemical sensors for the pharmaceutical, biomedical, and food industries [122].

## 4. Wearable and Flexible Electrochemical Glucose Sensors

Flexible and implanted glucose biosensors are a new technology for diabetes patients’ continuous blood glucose monitoring. It is necessary to develop flexible conductive substrates with a large active surface area in order to advance the technology. Zhao et al. prepared a free-standing non-enzymatic flexible electrochemical glucose sensor using a 3D monolithic nanoporous gold (Au) grown over a graphene paper which was further deposited with binary PtCo alloy nanoparticles using electrodeposition techniques. This flexible NPG/PtCo/GP electrode shows outstanding mechanical bending strength and can be integrated into a wearable biomedical or an implantable device. This flexible sensor offers great selectivity, repeatability, and stability, but it has a high manufacturing cost due to the many materials employed [125]. Similarly, another free-standing flexible sensor using graphene paper and binary PtCo alloy nanoparticles was reported by He et al. They formulated a conductive ink by combining 3D porous graphene-CNT assembly in ionic liquid (IL) and coating over this graphene paper. Due to the synergistic action of this 3D porous graphene-CNT assembly and PtAu alloy, this nanohybrid paper shows excellent sensing performances [126].

Abellán et al. created an enzymatic and non-enzymatic glucose sensor out of Pt-Graphite, which is regarded as the initial work on graphene-based wearable electrochemical glucose sensors [127]. Using simple, low-cost manufacturing procedures, Xuan et al. microfabricated and micro-patterned a reduced graphene oxide (rGO)-based nanohybrid working electrode on a flexible polyimide substrate. Here platinum and gold alloy nanoparticles were electrochemically coated over the microfabricated rGO surface containing glucose oxidase-chitosan. This sweat-based wearable sensor exhibits excellent amperometric response to glucose at a detection range of around 0–2.4 mM with a sensitivity of around 48 μA mM^−1^ cm^2^ and high linearity [128]. Cao et al. fabricated an electrochemical glucose sensor using a 3D paper microfluid that could prevent biomolecule immobilization from interfering with electrode modification. For glucose oxidase immobilization, an aldehyde-functionalized reference and counter-electrode hydrophilic zone were constructed, while rGO-TEPA/PB modified paper was used as a working electrode to quantify and detect H_2_O_2_ generated by coating Prussian blue over rGO-TEPA. Under ideal conditions, the proposed biosensor may be utilized to quantify glucose over a large linear range of 0.1 mM to 25 mM, with a detection limit of 25 µM [129]. Using a simple substrate-assisted electroless deposition (SAED) approach, a flexible non-enzymatic amperometric glucose sensor using laser-induced graphene decorated with a Cu nanoparticles (Cu NPs-LIG) composite has been successfully fabricated by Zhang et al. The Cu NPs-LIG sensor, as constructed, has a high glucose sensitivity of 495 µA mM^−1^ cm^−2^ with a low detection limit of 0.39 μM. Here electron transfers occur as a result of redox interactions between Cu NPs and glucose molecules; however, the Cu/graphene combination can speed up the electron transfer and improve the glucose-sensing efficiency. In this case, the continual addition of glucose leads to a stepwise current response. The response time of this sensor is quick, and it only takes 0.49 s to acquire a steady-state current. This composite’s large surface area aids in the fast diffusion of glucose molecules [130].

Similarly, another flexible non-enzymatic glucose sensor was fabricated by Lin et al. here, Cu-NPs were electrodeposited over a DVD-laser scribed graphene (LSG). This sensor shows more sensitivity than Zhang et al. reported with the Cu NPs-LIG sensor with a similar detection limit of around 0.35 μM. A significant limitation of reverse iontophoresis (RI)-based ISF extraction detection approaches is that ISF is significantly diluted prior to quantification due to its random extraction over a vast region of the skin, necessitating finger-stick validations. Lipani et al. recently devised a novel ISF extraction method that utilised an array of tiny pixels with graphene working electrodes decorated with Pt to produce a transdermal non-invasive glucose monitoring patch [131]. Park and colleagues created a wearable multifunctional sensor over a soft contact lens using hybrid architectures of 1D and 2D nanomaterials to provide wireless detection of intraocular pressure and tear glucose with improved resilience, flexibility, and conductivity. The main components were graphene and its hybrid with Ag nanowires, which were fabricated over a transparent and stretchy substrate to prevent any impact on the wearer’s eyesight ability. GOx was immobilized on the graphene channel via covalent bonding with the pyrene molecule via N-hydroxysuccinimide (NHS) chemistry. Both in vitro and in vivo real-time glucose detection were performed on a live bovine and a rabbit eyeball by tracking the increase in drain current as glucose concentrations increased [132]. A wearable diabetic patch made up of Au-doped graphene and a serpentine bilayer of Au mesh, as well as thermoresponsive microneedles, has demonstrated the feasibility of closed-loop wearable monitoring and feedback therapy on a single platform [133].

Even though these LIG-based electrochemical flexible/wearable sensors are popular still, their inherent electrical properties and low surface conductivity hinder their application in sensors [134]. To enhance the conductivity, Yoon et al. proposed a surface modification method using acetic acid. This simple treatment helps to increase the ratio of c–c bonds and thereby enhances the conductivity. Following this, a working electrode was fabricated by electroplating with Pt nanoparticles (PtNPs) and immobilising glucose oxidase using cyclic voltammetry. Finally, they used a real sample test to perform and evaluate the practical feasibility of an as-produced sweat glucose biosensor based on acetic acid-treated LIG electrodes [135].

Lei et al. developed a stretchy and wearable biosensor that uses MXene/Prussian blue (MXene/PB) to detect two metabolites in sweat, lactate, and glucose. The high conductivities and unusual structure of the MXene/PB composite significantly improved the electrochemical sensor’s performance over CNTs/PB and graphene/PB composites. One distinguishing aspect of this biosensor is that all active component of this sensor is self-contained and may be replaced at any time. The biosensor’s one-of-a-kind design ensures an adequate flow of O_2_ during the investigation, enhancing the long-term stability. Immobilizing lactate oxidase (LO) (for lactate) and GOx (for glucose) over porous and ultrathin CNTs/PB/Ti_3_C_2_T_x_/CFMs electrodes separately resulted in the working electrode. The electrodes were then inserted into the sensor platform’s slot. The pH sensor is made up of Ag/AgCl as the reference electrode and PANI as the working electrode. This sensor, which had been modified with glucose oxidase, demonstrated a linear dynamic response to glucose in artificial sweat spanning the range of 10 × 10^−6^ M to 1.5 × 10^−3^ M, with a LOD of 0.33 × 10^−6^ M. Similarly, this sensor modified with lactate oxidase demonstrated a linear relationship with lactate concentration spanning 10 × 10^−6^ M to 22 × 10^−3^ M, with a LOD of 0.67 × 10^−6^ M. The biosensor’s practical application was proved by analyzing three components in sweat during strenuous cycling. This wristband biosensor based on MXene gathers enough sweat in less than 2 min and independently detects the concentrations of lactate, glucose, and pH value. The pH sensor is essential for monitoring local pH levels. The selectivity of a wearable sensing device is crucial due to the multiplicity of interfering components found in human perspiration. Interestingly, lactate, AA and UA were shown to have no effect on the glucose sensor. Similarly, AA, glucose and UA had no effect on the lactate sensor’s response. This study shows how to make a wearable and flexible MXene-based electrochemical sensor for detecting human biomarkers, which can be used for advanced personal care and disease detection [136]. M. et al. created a highly integrated sensing (HIS) paper that used foldable all paper substrates and Ti_3_C_2_/MB as the active material to make sweat analysis patches. This HIS sensor contained a signal processing platform for sweat analysis that could detect lactate and glucose in real-time. Because of a well-designed 3D diffusion path and the hydrophilic action of paper substrates, the functional sections of the HIS sensor were printed on paper substrates and folded into a three-dimensional structure. Human perspiration could be effectively gathered and promptly diffused from human-device interfaces in a vertical manner. According to the HIS paper, they created a dual-channel electrochemical sensor that could detect lactate and glucose simultaneously with a sensitivity of 0.49 µA of 2.4 nA µM^−1^ and mM^−1^, respectively. The HIS paper’s low cost and convenience make it a good candidate for wearable bioelectronic devices and non-invasive electrochemical sensors [137]. Myndrul et al. developed a stretchable and skin-attachable enzymatic glucose sensor using ZnO tetrapods (TPs) and MXene very recently. Because of the superior electrical conductivity of MXene and high surface area of ZnO TPs, this electrochemical glucose sensor shows enhanced sensitivity in sweat samples with LOD around 17 μM in a broad linear detection range (LDR = 0.05–0.7 mM) and enhanced mechanical stability (up to 30% stretching) [138]. The employment of MoS_2_ and flexible polymeric materials as electrodes at the same time generates a synergetic effect for fabricating a flexible biosensor platform. In a study by Yoon et al., they fabricated a MoS_2_/Au NPs glucose sensor over a polymer electrode, where they fabricated this sensor using spin coating. This sensor shows flexure extension values around 3.36 mm and 3.48 mm, which is larger than that of a rigid electrode [139]. NiSe_2_ is a prominent TMD because of its higher electrochemical activity due to the synergistic effects of the two metals and the active sites present in them. Aside from that, it has a higher efficiency in electron transport, making it a good option for use as a biosensor. Vishnu et al. built a disposable glucose sensor using a simple hydrothermal process for direct NiSe_2_ development on cellulose paper [140]. Liu and colleagues used a simple hydrothermal process and a post-annealing treatment to create a 2D copper cobaltite (CuCo_2_O_4_) nanosheet with flower-like morphology decorated on a flexible graphite paper. Spinel cobaltite’s have higher electrical conductivity and electrochemical properties than monometallic oxides because electrons with low activation energy flow across multiple transition metal cations, resulting in better storage efficiency. Graphene paper with extra active sites has great mechanical strength, cheap cost, durability, and is lightweight, making it perfect for flexible electrodes [141]. Wang et al. fabricated another non-enzymatic sensor based on a 3D array of NiCO LDH over a carbon cloth using a one-pot co-precipitation method. Because of the loosely-packed fiber structure of carbon cloth, its high flexibility and superior conductivity, which promotes glucose diffusion at the electrode interface, CC is chosen as the conductive substrate to deposit the NiCO LDH active materials. Furthermore, the large surface area of the NiCO LDH NS and interconnected structure allows easy access to glucose molecules [142]. Another flexible non-enzymatic sensor based on NiAl-LDH modified carbon cloth was also fabricated by Hai and Zou [143].

To realize the miniaturized wearable glucose sensors, the currently employed energy sources are not sufficient. This constraint becomes even more significant in the case of miniaturized implantable devices with extended operation times, as the main impediment to fabricating these devices has been the large sizes of co-implanted electronic circuit boards and the batteries that power them [11]. The development of enzymatic BFCs as attractive self-sustaining energy devices that eliminate the need for external power sources by harvesting bioenergy directly from metabolites present in body fluids has been an appealing way to meet this critically significant demand. Willner and Katz published the first study on self-powered glucose biosensors in 2001 [144]. Cho et al. reported a self-powered BFC-based non-invasive glucose biosensor, which was incorporated into a regular Band-Aid adhesive patch to detect glucose in human perspiration. This biosensor was built up of chitosan/GOx/graphene-based bioanode and activated carbon (AC)/Ni-based cathode and polystyrene sulfonate:poly(3,4-ethylenedioxythiophene) (PSS:PEDOT) as a conductive reservoir. This skin-connected technology can immediately absorb sweat by capillary pressure into a conductive reservoir, which converts chemical energy into electrical energy and detects glucose without the need for an external power source [145].

## 5. Photoelectrochemical Sensors

Photoelectrochemistry, which originated from electrochemistry, is a vibrant subject that investigates the action of light over photoactive materials, involving the conversion of sunlight into electricity and the interconversion of chemical and electric energy [146,147,148]. Following the transfer of photogenerated charge carriers to the electrodes, electrical signals are formed in the electrolyte, and the energy conversion is performed by the associated redox reactions. Photoelectro chemistry has been widely employed in pollution treatment and energy harvesting because of these properties. Notably, the coupling of the PEC process with sensing has ushered in a novel but promising technology, PEC sensing. The photoactive materials at the interface of the electrode that serve as the signal converter in PEC sensing, produce an electrical signal (current mode is usually employed in current PEC sensors) that is influenced by light-irradiated targets. A basic sensing system is made up of three essential components: an excitation light source unit, a detecting system containing a metal electrode having electrical and catalytic activity, an electrolyte, and a working electrode tuned with photoactive materials, and finally, a signal reading device. Biological materials (nucleic acids, antibodies, enzymes, and so on) for particular recognition are also required for PEC biosensors. The creation of an electrical signal in the PEC detecting system as a whole requires a number of chemical and physical processes. The usually accepted mechanism consists of four parallel processes: (1) absorption of photons, (2) separation of charges, (3) migration and recombination of these charges, and (4) charge usage (producing an electrical signal and engaging in a redox reaction at the solid-liquid interface). Unlike traditional electrochemical sensing, the excitation source in PEC sensing is light, and an electrical signal is employed for readout. Because of the different energy forms between the excitation source and the detection signal, PEC sensing has the potential to be more sensitive than classic chemiluminescent and electrochemical techniques. Furthermore, unlike electrochemical sensing, which frequently generates signals at certain potentials, PEC sensing, which exploits the strong redox characteristic of electron-hole pairs, reduces reliance on the applied potential. As a result, when compared to electrochemical sensing with identical parameters, PEC sensing frequently beats electrochemical sensing in terms of sensing performance. As per the fundamental principles of photoelectric conversion, recombination is another key factor impacting the conversion efficiency of photoactive materials. As a result, many techniques for suppressing charge carrier recombination have been developed. The most common and effective methods are designing and constructing semiconductor–semiconductor heterojunctions, multicomponent heterojunctions and semiconductor–carbon heterojunctions because photogenerated carrier separation and transfer are more successful at the interface region of different components [149,150]. Multiwalled and single-walled carbon nanotubes, graphene oxide, reduced graphene oxide (rGO), and graphene hydrogel are the most popular carbon materials employed to fabricate the semiconductor–carbon heterojunction.

Graphene has the highest specific surface area of any carbon material and excellent chemical stability and electron mobility. As a result, it is regarded as a superior medium for electron transmission. Indeed, improving sensing performance by combining semiconductors with graphene benefits from a number of factors. Furthermore, a Schottky barrier junction between the semiconductor and graphene can be formed, increasing charge separation. On the other hand, graphene can enhance stimulated electron transport while inhibiting surface recombination due to its greater electron mobility. For example, in order to reduce the recombination rate and improve the photocatalytic capabilities of tungsten oxide (WO_3_), Devadoss et al. used CVD followed by sputtering to introduce graphene beneath WO_3_, and photocatalytic activity was further enhanced by depositing AuNPs. AuNPs not only improve enzyme catalytic activity but also improve charge collection at WO_3_/Graphene surfaces by producing a Schottky junction. Here the WO_3_ nanoparticles, which are photoactive, often form electron-hole pairs when exposed to light. These photogenerated holes were then scavenged by a biological analyte (which acts as an electron donor), resulting in the oxidation of the biomolecules via an intermediary step involving the electron acceptor FAD/FADH_2_ shown in Figure 10. At NHE of around—0.4 V, glucose is oxidized to gluconic acid throughout this cycle. Concurrently, photogenerated electrons convert water to create H_2_ on the cathode (Pt). The sensing signal is the produced photocurrent travelling across the circuit. Thus, the efficiency of the PEC sensor is solely dependent on the effectiveness of photoactive materials in responding to biomolecules of interest and transduction efficiency [151]. CdS is another photocatalytic material that is widely investigated in PES systems. Similar toWO_3_, this semiconducting material has a low recombination rate. To increase photovoltaic conversion efficiency and charge separation, CdS is modified with graphene oxide by Zhang et al. using the facile hydrothermal method. This hybrid was demonstrated as a photoanode material in non-enzymatic glucose sensors. Here rGO helps to stabilize the quantum dots using the functional group present in the rGO, and this composite shows enhanced photoelectrochemical properties [152].

Later Ma et al. electrochemically coated CoOx over these graphene oxide–CdS nanocomposites to achieve both electrocatalytic and enhance the photoelectrocatalytic oxidation of glucose. These CoOx/graphene oxide-CdS composites demonstrated remarkable performance as an enzyme-free photoelectrochemical glucose sensor, with a broad linear concentration range (0.005–0.37 mM), high sensitivity (796.7 μA mM^−1^ cm^−2^), and an LOD of around 0.5 μM with strong selectivity [153], better than the graphene oxide-CdS nanohybrid prepared by Zhang et al. (LOD around 7 μmoL dm^−3^) [152]. Another ternary hybrid based on graphene-CdS nanocomposites was reported by Jafari et al. They modified graphene oxide–CdS nanocomposites using electropolymerised Nile blue (P-NB). The produced nanocomposites displayed good electrocatalytic activity toward the oxidation of NADH at a relatively low anodic potential due to the synergistic action of P-NB and the graphene oxide–CdS nanocomposite (ca. 50 mV). The P-NB utilized in this study not only acts as a redox mediator for the electrocatalytic oxidation of NADH but also as an electron donor for the neutralization of photogenerated holes in CdS QDs, resulting in a considerable increase in response photocurrent. By covalently attaching the GDH enzyme with this nanocomposite, a photoelectrochemical sensor for glucose sensor was fabricated [154]. Laser-induced graphene that possesses fast charge transfer and ion diffusion employed as electrode materials for electrochemical energy storage devices motivated Li et al. to fabricate a photoelectrochemical glucose sensing. They created a Ni and CdS hybrid in LIG via a one-step laser-induced solid-phase transition. This PEC sensor exhibits outstanding photoelectric catalytic activity towards glucose, which is owing not only to the synergistic impact of Ni and CdS but also to the superior conductivity of the LIG’s 3D macroporous architecture. The electrocatalytic performance towards glucose and oxidation of glucose can be summarized by the equation given below:NiO(OH) + glucose → Ni(OH)_2_ + glucolactone(15)
NiS(OH) + glucose → NiS + glucolactone(16)

This laser-induced hybrid photoelectrode shows an LOD of around 0.4 μM with good selectivity, stability, and reproducibility [155]. Cuprous oxide (Cu_2_O) is a low-bandgap semiconductor with a tolerable bandgap value (2.0–2.2 eV) that opens up new avenues for increased solar light harvesting. It is also a low-cost, non-toxic semiconductor with improved stability. By combining it with n-type TiO_2_, this p-type semiconductor has been investigated for water photocatalysis. Using these p–n heterojunctions and the enhanced electron transport capabilities of rGO, Bekir et al. created a PEC glucose sensor based on Cu_2_O/rGO-coated TiO_2_ nanotubes (NTs) arranged titanium foil. The creation of p-n heterojunctions between TiO_2_ NTs and Cu_2_O NPs, as well as the good conductive rGO, can both improve charge separation efficiency and permit electron transmission in this case, resulting in significantly higher photocurrent [156]. Recently, a visible-light-driven PEC glucose biosensor was proposed by Zhao et al. by utilizing rGO/TiO_2_ nanotubes with GOD as the identification element. This constructed PEC sensor shows excellent sensitivity with an LOD of around 5 μM [157].

Aside from semiconducting materials, layered BiOXs (X = Br, Cl, and I) have garnered a lot of attention as a photocatalytic material. Among these, BiOCl is regarded as one of the most promising. BiOCl has a layered structure that is made up of [Bi_2_O_2_] 2p slabs separated by double slabs of chain atoms. BiOCl’s strong photocatalytic performance is attributable to its layered structure’s effective charge carrier separation capabilities, as well as the induced dipole formed inside its molecular structure. Combining the individual advantages of these BiOCl and graphene, Gopalan et al. used a hydrothermal method for the synthesis of 2D bismuth oxychloride–graphene nanohybrid sheets (BiOCl-G NHS). This nanohybrid formation takes place in three stages: (i) electrostatic adsorption of bismuth precursor onto carboxylic acid-functionalized graphene nanosheets, (ii) formation of BiOCl nanocrystals as nuclei on the surface of graphene nanosheets, and (iii) growth of BiOCl nanosheets on the surface of functionalized graphene sheets, resulting in stacked stacks of BiOCl and graphene nanosheets. The cyclic voltammetry and differential pulse voltammetry (DPV) studies demonstrated this nanohybrid is able to generate photocurrent for glucose when it is illuminated with a source of light having a wavelength of 365 nm. The photocurrents produced in the presence of glucose displayed a linear relationship on glucose concentration in the range of 0.5 to 10 mM at a bias potential of +0.50 V, with a detection limit of 0.22 mM [158]. Similar to BiOCl, another ternary bismuth semiconductor compound, BiOBr, was reported to fabricate a PEC sensor. A BiOBr-TiO_2_ nanotube array (TNTA) composite electrode was prepared using vacuum impregnation and the chemical precipitation method. Here the matched levels of BiOBr and TNTA enhance the separation of holes and photoelectrons effectively and significantly enhance the photoelectrochemical performance with LOD around 10 nM [159].

Graphitic carbon nitride (g-C_3_N_4_) is a relatively new form of carbon-based material that has just recently been synthesized. Due to its simple synthesis method, strong photo-stability, low cost, and adequate band gap, graphitic carbon nitride (g-C_3_N_4_) has attracted a lot of attention in PEC sensing. ZnIn_2_S_4_, a ternary chalcogenide semiconductor, has been widely used in photodegradation pollutants and photocatalytic hydrogen evolution. It should be noted that gC3N4 and ZnIn_2_S_4_ can create a heterojunction with an overlapping band gap structure, effectively separating photogenerated electrons and holes and increasing photoelectric conversion efficiency. Furthermore, g-C_3_N_4_ and ZnIn_2_S_4_ exhibit superior PEC response and photo-stability when exposed to visible light. As a result, g-C_3_N_4_/ZnIn_2_S_4_ composites for PEC sensing were fabricated by Zhang et al. Using Au nanoparticles; they co-assembled horseradish peroxidase (HRP) and glucose oxidase (GOx) over this hybrid. With the addition of glucose and 4-chloro-1-naphthol (4-CN), this electrode can effectively catalyze the oxidation of glucose into H_2_O_2_ and gluconic acid. Here HCP can speed up the oxidation of 4-CN and thereby reduce the precipitation reaction time. The deposit on the electrode surface has the potential to significantly diminish the photocurrent signal. Because of the signal magnification process, this bi-enzyme glucose sensor has a “signal-off” sensing mode for glucose detection, with a dynamic detection range of 1–10,000 μM and a low LOD of around 0.28 μM. In addition, the developed PEC bi-enzyme glucose sensor has enormous promise for quantitative glucose detection in practical applications [160]. In addition, Fe_3_O_4_/g-C_3_N_4_ [161] and WO_3_/g-C_3_N_4_ [162] heterojunction structures were also employed in the PEC sensing platform for glucose detection. Çakıroğlu et al. utilized g-C_3_N_4_, MnO_2_, and Au nanoparticles to modify the fundamental semiconducting properties of TiO_2_. p-n heterojunctions were formed between the interface of g-C_3_N_4_/TiO_2_ and MnO_2,_ which inhibit the recombination of excited electrons and also facilitate charge transport. Here the major role of g-C_3_N_4_ is to amplify the photocurrent by enhancing photogenerated electron-hole pairs and decreasing recombination [163]. When exposed to visible light, electron-hole pairs develop on MnO_2_, and the photogenerated electrons are quickly pushed from the conduction band (CB) of MnO_2_ to the CB of g-C_3_N_4_ [163].

MXene’s huge surface area and exposed metal sites enable more active sites for semiconductor growth. The very first MXene-based PEC sensor for glucose detection used a Ti_3_C_2_ MXene/Cu_2_O heterostructure. Cu_2_O, as a p-type semiconductor, had a low cathode photocurrent when exposed to visible light. Anode photocurrent was observed in pure Ti_3_C_2_. The cathode photocurrent increased dramatically after the combination because of the multilayered MXene’s superior high carrier mobility and metallic conductivity, which boosted hole transport from copper oxide’s valence band to MXene and accelerated the separation of photogenerated electron-hole pairs. However, when glucose was added to the buffer, the photocurrent decreased due to the partial reduction of the species being consumed. The Ti_3_C_2_ MXene/Cu_2_O composite demonstrated good PEC performance and sensitive photoelectric response to glucose with a LOD of 0.17 nM. [164]. Chen, G. et al. constructed a Z-scheme TiO_2_/Ti_3_C_2_/Cu_2_O heterostructure and used it as a photocathode in a self-powered PEC glucose sensor. An artificial indirect “Z-scheme” system was created by connecting two semiconductors (two separate photosystems (PS)) using electron mediators. According to their findings, the TiO2 nanoparticles were generated in situ on the MXene nanosheets by ethanol heat treatment of the MXene, and the Z-scheme of this nanocomposite heterojunction was prepared by hybridizing with the narrow bandgap semiconductor Cu_2_O. The Z-scheme of this nanocomposite heterojunction outperformed TiO_2_/Ti_3_C_2_ and Cu_2_O in terms of PEC performance, resulting in a TiO_2_/Ti_3_C_2_/Cu_2_O PEC sensor for glucose with a surprising low LOD of 33.75 nM and a broad range of 100 nM to 10 M [165].

TMDs and their composites are widely used in PEC glucose-sensing due to their advantages, including high charge mobility, narrow band gap, large surface area, and visible-light activity. MoS_2_ is the most popular 2D TMD in the PEC sector due to its ease of manufacture, low price, facile exfoliation, abundant raw ingredients, small band gap (about 1.8 eV), and visible light activity. Bulk MoS_2_ nanomaterials, on the other hand, are rarely used in PEC tests due to the considerable recombination of photogenerated electron-hole pairs. Several methods for modifying MoS_2_ in the PEC field have been developed to address this shortcoming, such as (1) exfoliation of bulk MoS_2_ to prepare single- or few-layered MoS_2_ nanosheets, (2) combination with other semiconductors to form heterojunction structures, and (3) modification with metallic nanoparticles. Using a simple thermolytical approach and a C_3_N_4_ sacrificial template, a 3D porous network made of ultrathin MoS_2_ nanosheets with good PEC performance was generated here. The following is the reaction procedure. The FAD redox group in GOD adsorbed on the MoS_2_ nanosheet surface oxidizes the glucose in the solution and transforms it to the reduced FADH_2_ group in this case. When exposed to visible light, the photo holes generated on the surface and inside the MoS_2_ films may easily donate holes to FADH_2_ and refill the enzyme redox centre FAD, while O_2_ molecules are reduced to H_2_O_2_. Electrons are transferred swiftly to the ITO substrate while being subjected to increased electrical power during this technique [166,167].

Liu’s group investigated the effect of TiO_2_ nanorods on the photocatalytic activity of MoS_2_ nanosheets and discovered that combining them with semiconductors helps to improve the photocatalytic activity of MoS_2_. The photoactivity of the MoS_2_-TiO_2_ composite was found to be significantly larger than that of pristine TiO_2_ and MoS_2_ nanorods. In fact, the photocurrent for this nanohybrid was 4.8 times that of pristine MoS_2_. It was explained as the matching band gap of MoS_2_ and TiO_2_, which promoted photogenerated electron transport to the electrode surface while inhibiting photogenerated electron-hole pair recombination. It was used to build a PEC biosensor by immobilising glucose oxidase (GOx) on a MoS_2_-TiO_2_/ITO surface. The photocurrent of this sensor grew linearly as the concentration of glucose increased from 0 to 8 Mm [168,169].

The other class of two-dimensional materials that were recently explored for glucose sensing were transition metal nitride materials. They are of interest due to their great electrical conductivity, biocompatibility, thermal and chemical stability, and electronic structural similarities to valuable metals. Some of the other reported electrochemical glucose sensors based on transition metal nitride and other 2D materials are given in Table 4. [170,171,172,173,174,175,176,177]

## 6. Theoretical Perspective for Glucose Sensing by 2D Materials

Experimentally, the dearth of speedy and scrimping synthesis procedures and unavailability of pre-testing facilities to confirm its proper functioning makes it challenging to develop and devise glucose sensors based on novel 2D materials. At present, various synthesis approaches such as CVD, sputtering, hydrothermal, electrodeposition methods etc., are being employed to fabricate fine-quality 2D materials for biomolecule detection. Nevertheless, most of these methods are costlier and cannot be used for mass production. Indeed, such constraints can be overwhelmed if the sensor materials are designed, modelled, and simulated using ab-initio theoretical techniques [178]. The ab-initio, also known as the first principal methods, are quantum mechanical methods that compute the electronic structure of the system by solving the Schrodinger equations numerically. Density Functional Theory (DFT) [179], Monte Carlo [180], Hartree, Fock [181] etc., are the standard quantum methods. DFT is the most effective and widely used technique, which proffers quick and precise determination of ground-state electronic properties of various materials. Unlike the typical laboratory methods, theoretical simulations have an explicit gain in both time and endeavor. The computational simulations utilizing DFT could produce atomic level clarifications and comprehensions of the laboratory results or even act as a predictive aid for designing novel sensors.

In the glucose-sensing process, the interaction between the glucose molecule and the 2D substrate materials is purely weak. From the atomic level perspective, in addition to understanding the adsorption mechanism, the fundamental investigation of the interaction between the glucose molecules and the surfaces of the 2D materials also provides a theoretical insight into hindering the glucose adsorption, which has essential practical relevance and research value. After the adsorption process, the surface of the 2D substrate material varies significantly. It is challenging to control the formation of point defects and clarify the grassroots mechanism experimentally. Hence, the theoretical simulations and computational calculations provide intimate knowledge of glucose sensing that cannot be traced barely from the experiments, namely optimum adsorption configurations, active sites, binding energy, variations in the electronic and optical properties of the host material after adsorption, bonding process, charge transfer (both amount and direction) etc. As the theoretical investigations on glucose-sensing by 2D materials are very scarce, these prolific details will encourage the prediction of the glucose-sensing properties of novel 2D materials, in turn, unwrap new tracks for the experimentalists to develop functionalized sensor materials with high sensitivity and selectivity.

### 6.1. Theoretical Modelling

Modelling the glucose adsorption system is of prime importance in examining the glucose adsorption mechanism. Since the molecule determination is a surface phenomenon, a 2D model of the substrate material satisfying the planar periodic boundary condition must be considered. The lattice parameters, bond distances and angles of the modelled structure should be consistent with their actual values. The interlayer interactions can be eluded by inserting enough vacuum spacing vertically above the 2D substrate materials. The next step is to optimise the modelled 2D material to a minimal energy configuration. This can be achieved by choosing suitable energy thresholds and k-point mesh samples for Brillouin zone integration. For 2D materials, the K-point mesh should have a unit value in the Z direction, such as 2 × 2 × 1, 5 × 5 × 1, 11 × 11 × 1, etc. Figure 11a,b shows the supercell of the 2D novel graphyne and Penta–Octa–Penta (POP)-graphene. [182,183].

Similarly, the structure of the glucose molecule should also need to be modelled, and each atom in the glucose should be optimized to its ground state configuration with minimum residual forces between them. Suppose the software used for implementing the DFT calculation is periodic. In that case, the glucose molecule should be enclosed in a large unit cell with proper vacuum spacing before the optimizations to evade the effect of periodic interactions (see Figure 11c). The glucose molecule is then placed over the surface of the 2D materials at different positions in different orientations. The final structures are relaxed to minimum energy state configurations. The adsorption energy calculation decides the most stable or favorable adsorption configuration.

### 6.2. DFT Simulation of Glucose Sensing by 2D Materials

The literature presents various explications of the glucose-sensing behavior of 2D materials. From the theoretical stance, the glucose molecules are physisorbed on the surface of the 2D nanomaterials. In this case, the glucose interacts with a long bonding distance, weak binding energy, and negligible charge transfer. The molecule adsorption usually induces modifications in the optical and electronic properties of the host materials. Whereas during the physisorption, the electronic and optical properties remain nearly constant. This is because of the weak hydrogen bonding or van Der Waals interactions between the 2D materials and the glucose molecules.

Generally, physisorption occurs when the pristine 2D materials are adopted as the substrate material for molecule adsorption. The introduction of defects, doping with foreign materials, heterostructure formation, etc., create more active sites on the surface than its perfect structure [185,186,187,188]. In addition, the tuning can cause a sudden change in the electronic states of the 2D material, which promotes the sensing mechanism [189]. For example, the pure MoS_2_ monolayer is a semiconductor with a direct bandgap of about 1.86eV. The substitutional doping of Nb into MoS_2_ lowered the Fermi level down to the valence band, implying a substantial improvement in conductivity, as shown in Figure 12c–e [189]. In effect, the glucose molecule can form a strong chemical or covalent bond with the surface of the defected or functionalized 2D materials. Such a type of adsorption process is called chemisorption. Here, the glucose molecules bind to the surface with strong adsorption energy and large charge transfer. As a result, the electronic properties of the host 2D material drastically vary immediately after the adsorption process.

The glucose sensing performance of the 2D materials can be predicted theoretically by analyzing the following parameters:i.***Adsorption sites and configuration***

Depending on the geometry, 2D materials have different adsorption sites with which the glucose molecules can interact. For example, various adsorption sites of graphyne and POP-graphene are indicated in Figure 11a,b. The glucose molecule can adsorb at various adsorption sites in different configurations. Different configurations imply that certain atoms of the glucose molecule, such as O, C, and H, are placed adjacent to various adsorption sites. Secondly, the glucose molecule can be oriented both parallelly or perpendicularly to the surface of the 2D material. Before geometrical optimization, a reasonable distance between the glucose molecules and the 2D surface should be set for each adsorption configuration. The adsorption energy and bond distances between the 2D material and the glucose molecule decide the most favorable adsorption sites or stable configurations. The more negative adsorption energy, the higher the configuration’s stability will be. Therefore, the structure with the lowest adsorption energy is identified as the optimal configuration. Figure 13a,c and Figure 14a,b illustrate the geometries of glucose adsorption in a pristine 2D C_2_N system and metal-doped graphene, respectively.

ii.
**
*Adsorption energy*
**


The adsorption energy defines the strength of the interaction of the glucose molecule with the 2D substrate materials and can be determined by the following formula:(17)$$E_{\mathrm{Ad}}=E_{2D\;\mathrm{substrate}\;+\;\mathrm{glucose}}-\left(E_{2D\;\mathrm{substrate}}+E_{\mathrm{glucose}}\right)$$

Each term in Equation (17) represents the energy of the corresponding optimized structures before and after the adsorption. The glucose molecule adsorbs spontaneously on the 2D material surface if the predicted adsorption energy values become negative. On the other hand, the positive result indicates that the adsorption process is endothermic, implying that the 2D material under investigation lacks the glucose-adsorption property. The adsorption energy can vary with the adsorption sites. The adsorption strength is proportional to the magnitude of the adsorption energy value. However, a significantly higher value may impact the sensor’s utility and reusability. If the adsorption energy of glucose on pristine 2D materials is poor, numerous tuning techniques can be employed to improve the adsorption ability. The standard LDA and GGA exchange functionals exclude long-range interactions when computing the adsorption energy [190,191]. Therefore, specific dispersion corrections, such as Grimme’s DFT-D2, D3 etc., are included during the simulation [192,193]. The adsorption energies of glucose with various 2D materials are listed in Table 5.

iii.
**
*Variations in the electronic properties*
**


The electronic properties of the pristine 2D materials alter soon after the adsorption of glucose molecules. The electronic properties will change significantly if the glucose molecules are chemisorbed on the 2D substrate. These modifications can be explained by comparing the electronic band structure and the total density of states (DOS) plots of the 2D material before and after the glucose adsorption. Glucose adsorption may induce new energy states near the Fermi level. If the 2D material used as the substrate is semiconducting, the adsorption can cause a reduction in the bandgap. For example, compared to the DOS of the pristine C_2_N monolayer in Figure 13b, the energy states near the Fermi level are enhanced after the glucose adsorption [195]. However, the bandgap remained the same even after the adsorption (Figure 13d).

iv.
**
*Charge transfer*
**


The surface of the 2D material behaves either as charge donors or acceptors. When the glucose molecule interacts with the 2D material, they are adsorbed on the surface by transferring charges. Superior charge transfer indicates the high sensitivity of the 2D material. The DFT analysis aids in the interpretation of the binding mechanism by examining the charge transfer between the glucose molecule and the 2D material. The HOMO and LUMO plot of D-glucose in Figure 11d,e reveal the electron-accepting nature of D-glucose [184]. The quantitative measurement of the charge transfer can be determined by different tools such as Bader charge analysis, Mulliken charge analysis [196], Lowdin charge analysis [197], etc., and is expressed by equation 18.
(18)$$∆q=q\left(2D\;\mathrm{material}+\mathrm{glucose}\right)-q\left(2D\;\mathrm{material}\right)-q\left(\mathrm{glucose}\right)$$
(19)$$∆\rho =\rho \left(2D\;\mathrm{material}+\mathrm{glucose}\right)-\rho \left(2D\;\mathrm{material}\right)-\rho (\mathrm{glucose})$$

If ∆q is positive in equation 2, charge transfer occurs from the glucose molecule to the 2D material and vice versa. The amount of charge transferred between the glucose and various 2D material systems is listed in Table 5. The charge gained by the glucose molecule is due to its electron-withdrawing property, as indicated by the negative ∆q in the table. Moreover, the charge flowing direction during the adsorption process can be assessed from the charge density difference (Cdd) plots. Equation (3) gives the mathematical description of the Cdd, where ρ is the density. The Cdd of the D-glucose adsorption on N-, O-, and Cl-doped HGr is shown in Figure 14d–f. The blue color enveloping the glucose molecule at the dopant site indicates that the molecule receives electrons from the doped hGr. Furthermore, the Cl-doped HGr interacts strongly with the glucose molecule leading to a large charge transfer of around −0.20|e| from the system to the D-glucose molecule [184].

v.
**
*Orbital interactions*
**


DFT investigations can be used to study the orbital level binding mechanism. As previously stated, charge transfer is a key factor in glucose sensing; the qualitative concept of charge transfer across atomic orbitals can be explored by analyzing the partial density of states (PDOS). The *2p* orbital of the glucose molecule interacts with the metal *d* orbitals via *p–d* hybridization if the 2D material under discussion is TMDs, MXenes, or any other transition metal-based structure. The strong orbital hybridization between the C *2p* and X *3p* orbitals near the Fermi level in X-doped HGr (X = Cl, N and O) is the reason for the increased glucose adsorption (see the PDOS in Figure 15a–c). The Cl *3p* orbitals have enhanced peaks and large overlap with the O *2p* orbitals of glucose molecule near the Fermi level than the N and O *3p* orbitals, explaining its large adsorption activity [182].

### 6.3. Theoretical Analysis of the Practical Feasibility of the Glucose Sensor

For the convenient operation of the proposed glucose sensor, the material should be stable, sensitive, selective, and reusable. This section details how to theoretically assess the potential of 2D material-based glucose sensors.

(i)
**
*Thermal stability*
**


Since the DFT calculations are performed at 0 K, it is essential to check the room temperature stability of the system comprising the sensing device. Ab initio Molecular Dynamic simulations (MD) at 300 K are undertaken to ensure the same [198,199,200]. In addition, the reuse of the sensor device requires that glucose molecules be readily absorbed from its surface, which often occurs at a temperature higher than room temperature. Yong et al. performed the first principal MD calculations at different temperatures to estimate the thermodynamic stability of the B_6_N_6_H_6_ monolayer [201]. They found that at 1000 K, the total energy of the monolayer shows a stable oscillation around a mean value (see Figure 16), but at 1200 K, the system starts to decompose and completely decomposes when the temperature reaches 1500 K. The MD snapshots of the geometric structure at these three temperatures are shown in Figure 15d–f. The excellent thermodynamic stability below 1000 K allows the B_6_N_6_H_6_ structures to act as potential room temperature sensors. 

(ii)
**
*Sensitivity of glucose adsorption analysis based on the band structure*
**


As aforementioned, glucose adsorption alters the inherent bandgap inherent electrical bandgap (EG) of the semiconducting 2D material. The study of this variation of bandgap helps in analysing the changes in the electrical conductivity (σ) of the glucose adsorption system using the following equation:(20)$$\sigma \propto \exp\left(\frac{-E_{G}}{2\mathrm{KT}}\right)$$

Here K and T, respectively, denote the Boltzmann constant and operating temperature of the glucose sensor [200]. The higher the value of σ, the greater the sensitivity of the system. A significant reduction in the bandgap improves the conductivity of the material, which can be acquired as an electrical impulse via the external circuits.

The bandgap variation after the glucose adsorption can sometimes be very low; therefore, it should be computed precisely. The correctness of the DFT results is highly influenced by the exchange correlational functional chosen. The widely used LDA and GGA are insufficient to define the localised electronic states in TMs and other elements with higher atomic numbers. We need to incorporate some corrections to DFT to address the shortcomings of standard GGA and LDA methods such as DFT + U [202], hybrid functionals such as the HSE06, B3YLP, and GW approximations [203,204,205], etc.

(iii)
**
*Recovery time*
**


The time taken for the desorption of glucose molecules from the surface of the 2D material substrate is called the recovery time (τ); or in other words, it defines the time taken by the sensor material to retain its original state after the desorption process. A quick recovery of the sensor material is vital for better functionality; otherwise, the sensor will be termed disposable. It has a strong correlation with the temperature and the potential desorption barrier. The adsorbed system is frequently heated to high temperatures to accomplish recovery. A theoretical prediction of the recovery time can be made based on transition-state theory [206]:(21)$$\tau =A^{-1}\;\exp\;(\frac{\left|\mathrm{Eb}\right|}{\mathrm{KT}})$$
where A, E_b_, K, and T illustrate the attempt frequency, desorption barrier, Boltzman constant and temperature. Since the desorption is assumed as the opposite process of adsorption, the E_b_ can be made equal to E_Ad_. If the adsorption of the glucose molecule on the 2D surface is stronger, the more extensive will be the recovery time. The reusability of the glucose sensor may be hindered by prolonged recovery periods.

## 7. Future Perspectives

Two-dimensional materials emerge as the advanced and high-performance electrode for glucose sensing application, but several challenges still remain that need to be addressed in future.

(1)Electron transfer between the current collector and the catalytic active sites is governed by the electrical conductivity of the catalyst. The use of conductive materials can promote interfacial contact by lowering interfacial charge resistance and overpotential. Because of their great charge mobility and strong electrical conductivity, 2D materials can be employed to accomplish this. High stability, little residual current, high capacitance, vast flexibility, and substantial robustness characterize 2D-material-based electrochemical glucose sensors.(2)Another important quality of a glucose sensor is judged by the stability, which is reliant on a number of parameters. Graphene and BN have the highest chemical and structural stability of any 2D material, whereas other 2D layered materials can react with water and oxygen. Black phosphorus (BP) has a high reaction rate when compared to other stacked 2D materials. As a result, publications based on BP for glucose detection are relatively rare. Gu et al. recently described an enzymatic biofuel cell based on BP and explored the BP mimicking enzyme properties [207]. Although certain physical and chemical modifications can be made to improve the stability of 2D materials, they nonetheless present numerous barriers and challenges. One major challenge is that existing solutions for enhancing the stability of 2D-material-based devices are still in the early stages of commercialization and manufacture [208].(3)Since glucose monitoring is connected with direct contact with real samples such as blood, urine, serum, and so on, the use of a biocompatible catalyst that inhibits biological reactions and cell damage is critical. Based on synthesis technique, structure, layer number and elemental composition, the compatibility of 2D materials may vary. For example, Tao et al. studied the biocompatibility of TMDs and found out that WSe_2_ shows high toxicity compared to WS_2_ and MoS_2_, which can be attributed to the toxic effects of Se. Similar studies were used to examine the role of the chalcogen atom in biocompatibility and found that VS_2_ is less toxic compared to VSe_2_ and VTe_2_. The biocompatibility of graphene and analogues has already been well-studied and find their place in glucose sensors. Aside from graphene and TMDs, the biocompatibility of additional 2D family members such as h-BN, MXenes, and black phosphorus is understudied. MXenes are quickly finding a place in healthcare applications; certain studies proved their biocompatibility. However, the biocompatibility of these electrochemical glucose sensors based on 2D materials should be further explored because in vivo devices must meet more stringent biocompatibility requirements than in vitro diagnostics.(4)The accurate detection of glucose in complicated physiological media (e.g., blood, interstitial fluid, or saliva) is critical for commercially viable biosensors, including those based on 2D materials. Because of the presence of non-target species in these media, there is an additional difficulty known as biofouling. When exposed to a biofluid, undesired molecules and other species (e.g., proteins, bacteria, platelets, etc.) might build on the sensor’s surface, interfering with and deteriorating its capabilities. Surprisingly, 2D materials can have antifouling capabilities by repelling hydrophilic interferents and so maintaining stability. In general, the electrooxidation of poisoning intermediates results in a large reduction in active surface area, which reduces sensitivity. Because of the huge surface area and edge-plane defect sites in 2D materials, this constraint can be efficiently eliminated. This needs a thorough assessment of the performance of these 2D-material-based biosensors in real-world matrices (beyond PBS or other buffers).(5)The use of 2D materials in wearable devices, particularly sweat monitoring platforms, has recently gained popularity. Such 2D-based platforms have demonstrated promising performance in sensing and self-powered energy harvesting devices. The use of cost-effective fabrication techniques (e.g., screen printing), improved reproducibility and accuracy of the measured glucose signal, system integration in body-compliant flexible wearable platforms, validating sensors with blood glucose levels, and the improved (stimulated) collection of non-invasive biofluids have all resulted in significant advances. Despite these exciting recent improvements, properly deploying non-invasive glucose monitoring systems requires additional tuning followed by large-scale validation studies to critically analyse the reliability and accuracy of these devices. These lengthy investigations will establish the potential of such non-invasive approaches to dependably monitor glucose and the association of these non-invasive generated data with gold-standard blood glucose readings. Other issues that need to be addressed in epidermal non-invasive sensing include irregular biofluid extraction, surface contamination, and the effect of some physiological factors (e.g., temperature, pH) on measurement accuracy. Despite these obstacles, the next decade will see a surge in activity in the development of non-invasive glucose monitoring systems based on 2D materials.(6)The incorporation of 2D materials into various semiconductor-based photoactive materials might increase their PEC sensing performance. Despite significant advances, there is still a long way to go in the research and implementation of 2D photoactive materials. Two-dimensional materials with a suitable band gap and strong optical absorption such as BP and 2D perovskites have not yet been reported. The lack of a suitable synthetic method also challenges the use of 2D materials in PEC sensing. However, the flexibility of ultrathin nanosheets, along with their strong photoelectric characteristics, makes 2D materials ideal for developing wearable PEC sensors for non-invasive health detection.

The future perspectives of these 2D-material-based glucose sensors are provided in Figure 16.

## 8. Conclusions

In this review paper, we examined recent advancements in enzymatic and non-enzymatic glucose sensor applications. Both types of glucose sensors are already taking advantage of the fascinating features of 2D materials on a regular basis. Considering their high sensitivity and specificity, enzyme sensors are sensitive to a narrow range of temperatures, pH levels, and humidity levels, which has sparked interest in non-enzymatic electrochemical glucose sensors. The activity of the majority of non-enzymatic glucose sensors is a direct function of the electrode material on which the glucose is oxidized. According to the study, the majority of glucose sensors’ catalytic activity is mostly dependent on the electrode material, which serves as the site of glucose oxidation. Earlier studies concentrated on elements such as Ag, Au, Pd, Cu, Pt, and Ni, as well as metal oxides such as iron oxide, silver oxide, copper oxide, and nickel oxide, among others. Researchers were particularly interested in metal/metal oxide compounds/alloys and metal oxide composites during the time. Polymers have recently generated interest due to properties such as biocompatibility and excellent selectivity.

## Figures and Tables

**Figure 1 biosensors-12-00467-f001:**
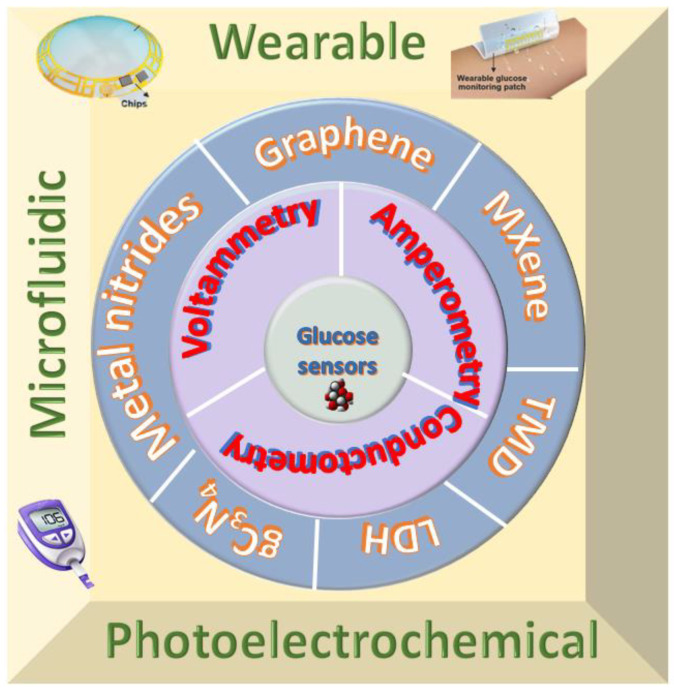
Summary of recent advancement of two-dimensional materials for electrochemical glucose sensors.

**Figure 2 biosensors-12-00467-f002:**
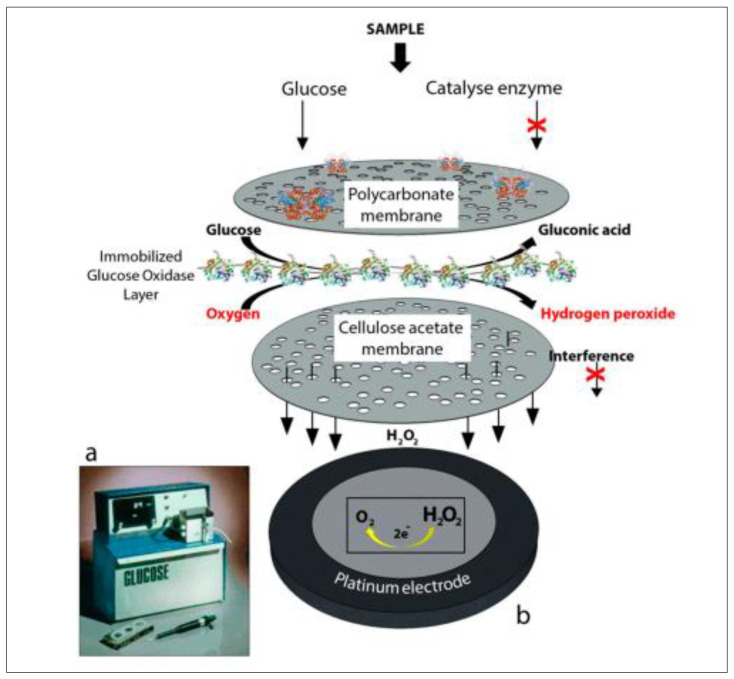
(**a**) YSI 23A Bio-Glucose Sensor. Reprinted with permission from Ref. [26]. 2003, Elsevier (Amsterdam, The Netherlands). (**b**) two-membrane system for glucose detection. Reprinted with permission from Ref. [27]. 2005, Elsevier.

**Figure 3 biosensors-12-00467-f003:**
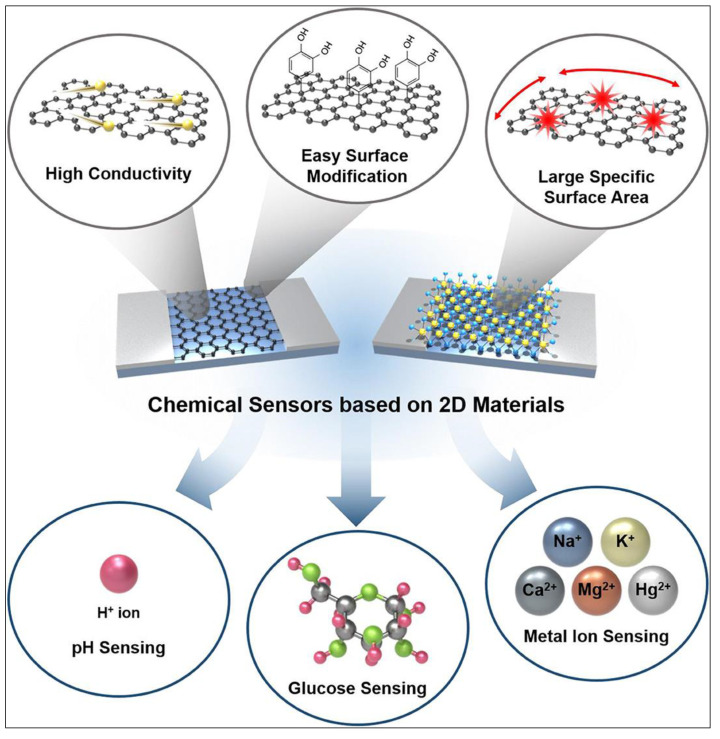
Schematic illustration of chemical sensors based on 2D materials with advantages of 2D materials and their application into ion/molecule sensing. Reprinted with permission from Ref. [41]. 2019, Lee, Suh and Jang.

**Figure 4 biosensors-12-00467-f004:**
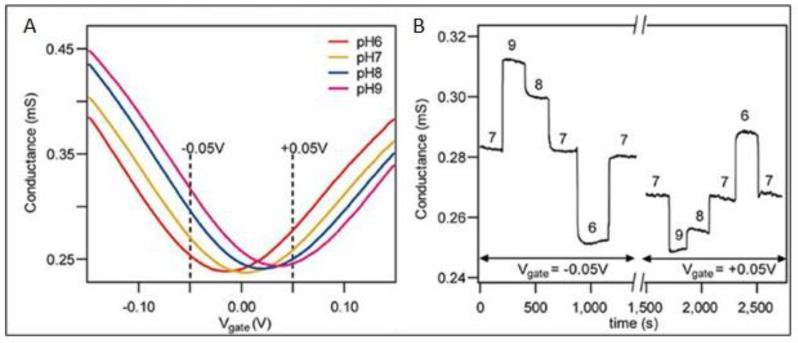
Glucose detection performance with (**A**) CVD Graphene FET and (**B**) response curves of glucose to different gate voltages. Reprinted with permission from Ref. [47]. 2012, Elsevier.

**Figure 5 biosensors-12-00467-f005:**
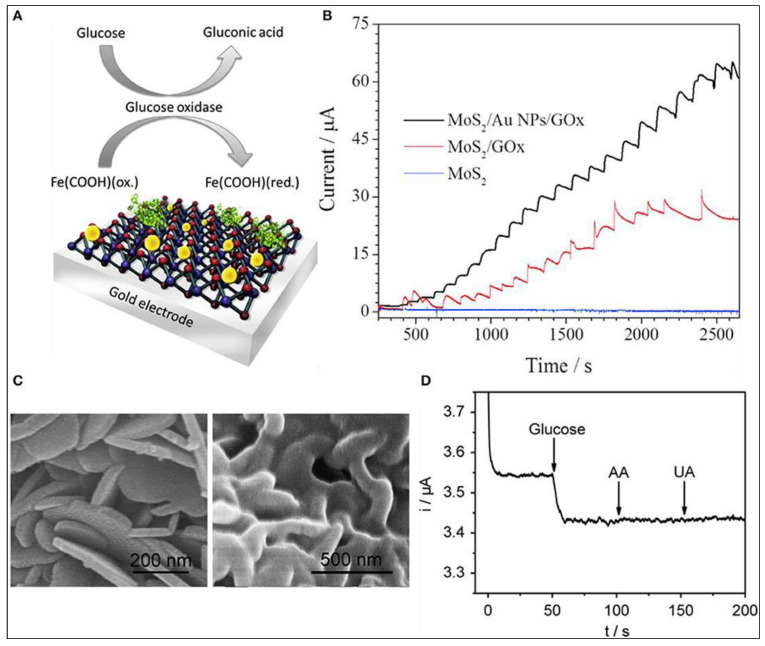
Glucose sensors using transition metal dichalcogenides. (**A**) Illustration for basic glucose sensor using Au nanoparticle decorated MoS_2_. (**B**) Response comparison between pristine MoS_2_, MoS_2_/GOx, and MoS_2_/Au NP/GOx electrodes. Reprinted with permission from Ref. [48]. 2017, Elsevier. (**C**) Scanning electron microscope images of pristine SnS_2_ (**left**) and GOx/MWCNTs–SnS_2_ (**right**). (**D**) Response curves of glucose having selectivity at GOx/MWCNTs–SnS_2_. Figures (**C**,**D**) were (Reproduced from [50]. 2014, Elsevier).

**Figure 6 biosensors-12-00467-f006:**
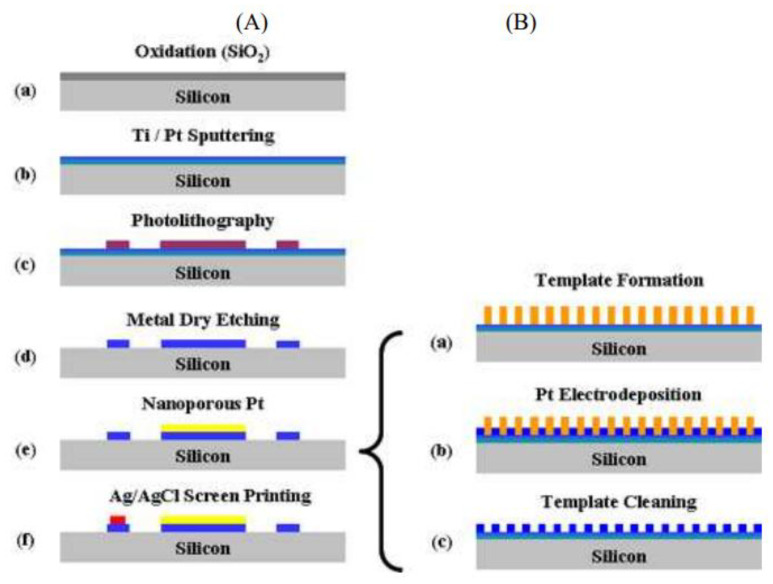
Fabrication procedures of non-enzymatic glucose micro-sensor with nanoporous Pt WE (**A**) and nanoporous Pt electrode (**B**) on silicon substrate. Reprinted from Ref. [58].

**Figure 7 biosensors-12-00467-f007:**
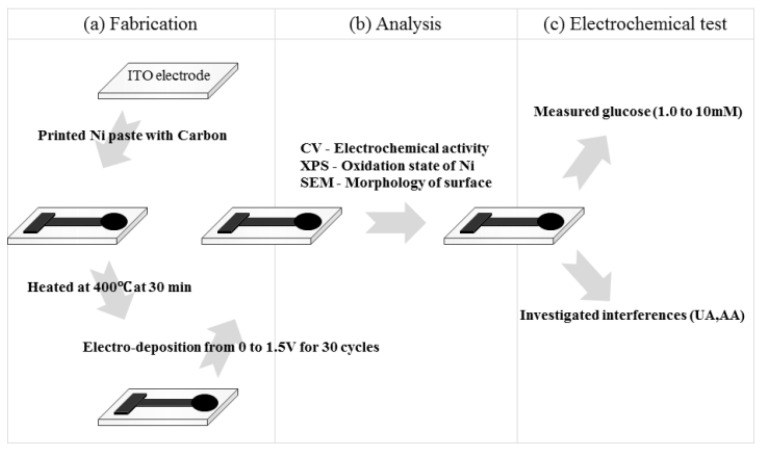
Schematic illustration of the (**a**) fabricating of Ni(OH)_2_/NiOOH-DSPNCE; (**b**) analysis of Ni(OH)_2_/NiOOH-DSPNCE; and (**c**) the electrochemical test. Reprinted from Ref. [59].

**Figure 8 biosensors-12-00467-f008:**
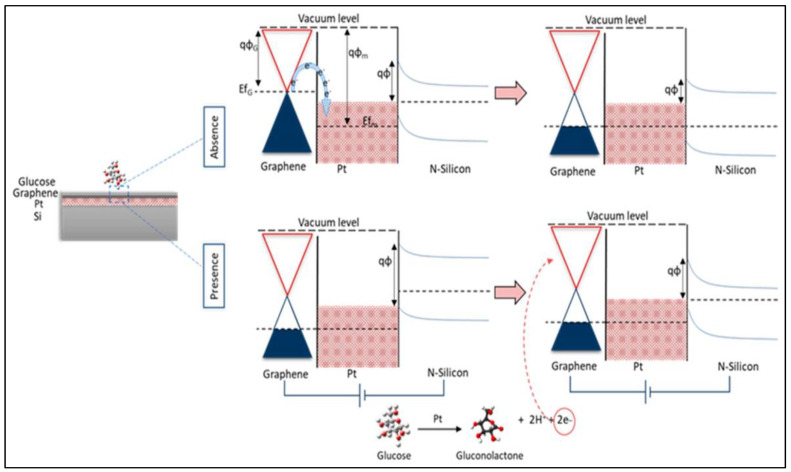
Graphene (G)/platinum oxide (PtO)/n-silicon (Si) heterostructure based glucose sensor working principle. Reprinted Ref. [70].

**Figure 9 biosensors-12-00467-f009:**
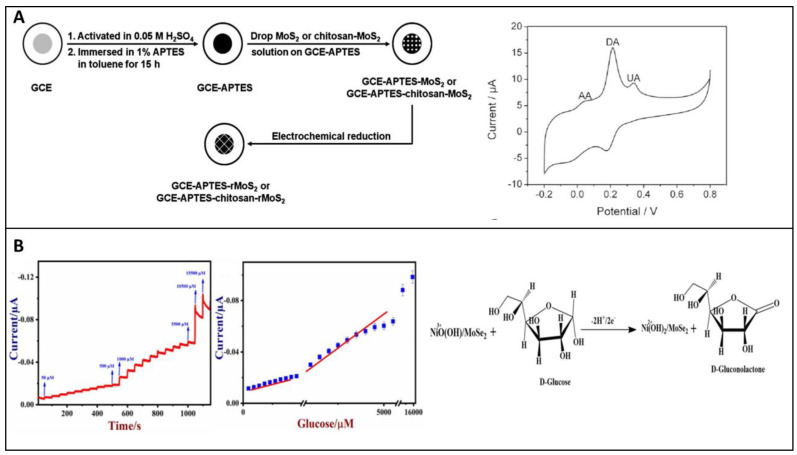
(**A**) Steps involved in the preparation of electrochemically rMoS_2_ and chitosan modified MoS_2_ electrodes and its CV curve in 0.1 M PBS. (Source: Reprinted with permission from Ref [100] *Small* 2012, 8, 2264–2270, doi:10.1002/smll.201200044. from Elsevier) (**B**) Amperometric curve NiO/MoSe_2_/GCE with the successive additions of glucose and its oxidation mechanism. (Source: Reprinted with permission from Ref [101] Scientific Reports Available online: https://www.nature.com/articles/s41598-021-92620-2 (accessed on 24 April 2022) from Elsevier).

**Figure 10 biosensors-12-00467-f010:**
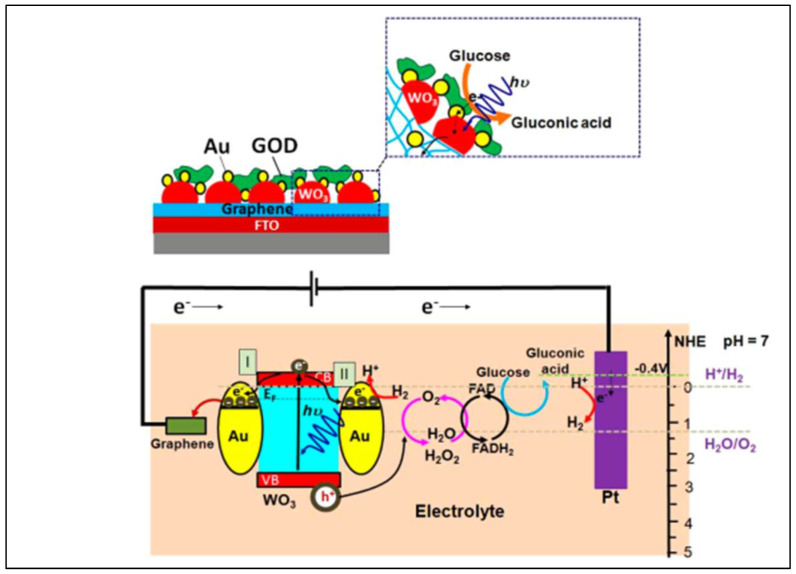
Schematic diagram of Graphene-WO_3_-Au triplet junction and its energy levels. Reprinted with permission from Ref [152]. *Electrochimica Acta*.

**Figure 11 biosensors-12-00467-f011:**
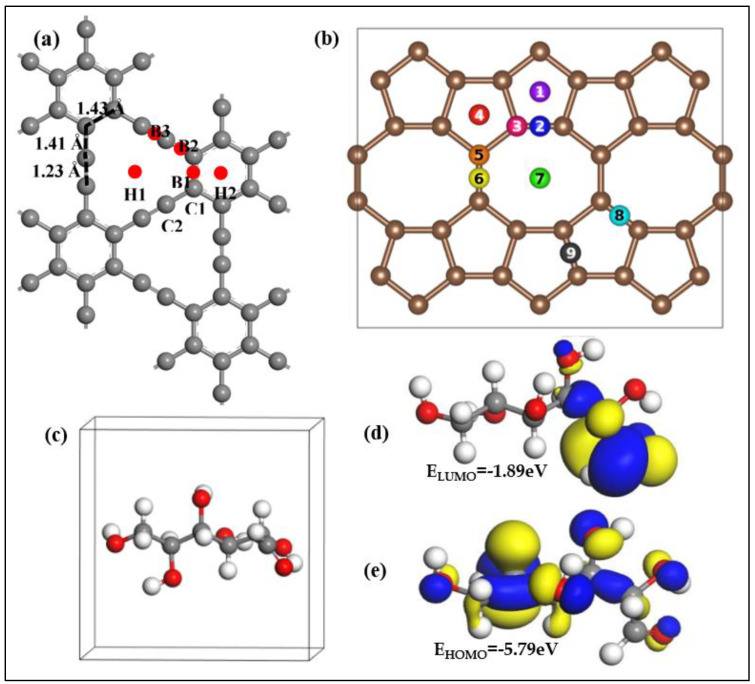
DFT optimized structures of (**a**) Graphyne, Reprinted from Ref. [182]. (**b**) POP-graphene with various adsorption sites, Reprinted from Ref. [183]. (**c**) D-glucose molecule and (**d**,**e**) HOMO and LUMO plots of D-glucose molecule. Reprinted from Ref. [184].

**Figure 12 biosensors-12-00467-f012:**
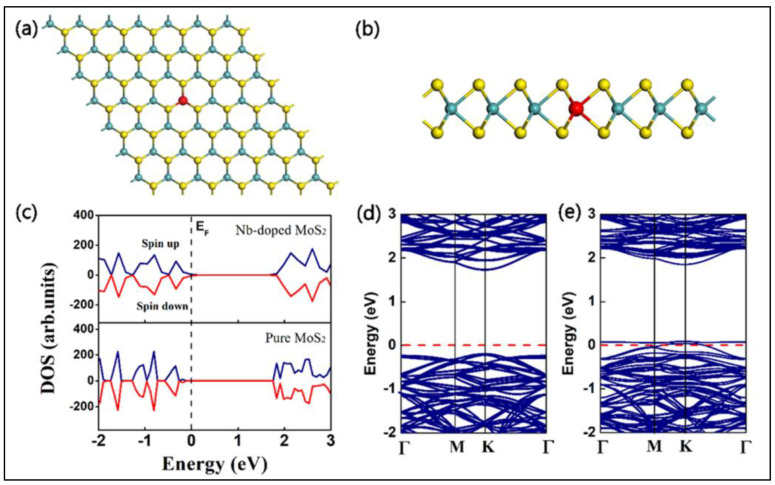
Geometries of 7 × 7 supercell of Nb-doped MoS_2_ in its (**a**) top and (**b**) side view. (**c**) Comparison of the total density of states plot of Nb-doped (upper panel) and pristine MoS_2_ (lower panel). Band structure plots of (**d**) pristine MoS_2_ (**e**) Nb-MoS_2_, Reprinted from Ref. [189].

**Figure 13 biosensors-12-00467-f013:**
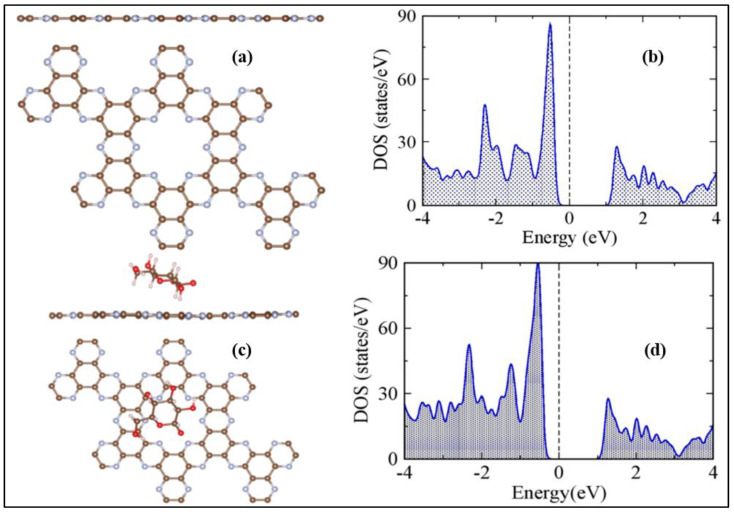
Optimised geometries and DOS of (**a**,**b**) 2 × 2 × 1 C_2_N monolayer (**c**,**d**) Glucose adsorbed C_2_N. Reprinted from Ref. [195].

**Figure 14 biosensors-12-00467-f014:**
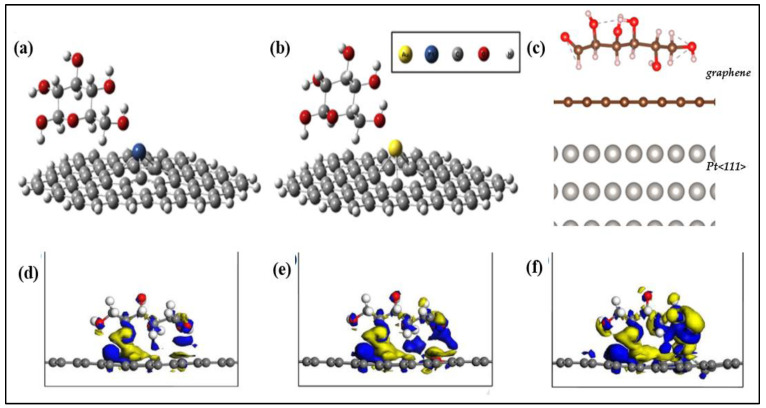
DFT optimised structures of glucose interaction with (**a**) Ag- (**b**)Au-doped graphene (**c**) Pt/graphene heterostructure. Reprinted with permission from Ref. [194], 2020, Elsevier. Charge density difference plots of glucose adsorbed (**d**) N- (**e**) O- and (**f**) Cl- doped hydrogenated graphene (HGr). Yellow and blue surfaces represent the charge density reduction and gain, respectively, with an isovalue of 0.0075 e Å^−3^. Reprinted from Ref. [184].

**Figure 15 biosensors-12-00467-f015:**
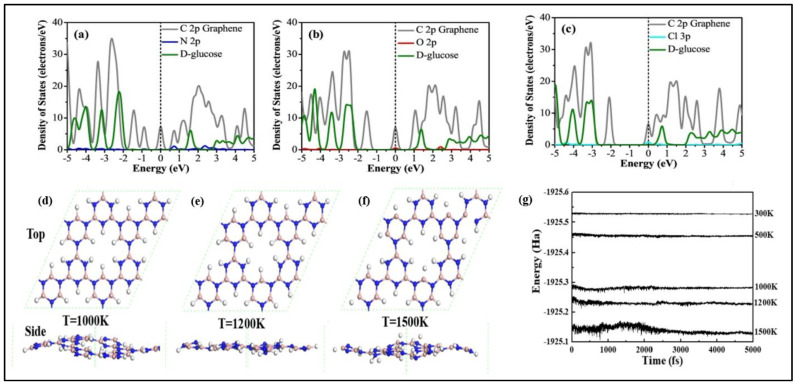
PDOS plots of D -glucose molecule adsorption on the (**a**) N-, (**b**) O-, and (**c**) Cl- decorated HGr, Reprinted from Ref. [184]. (**d**) 1000 K, (**e**) 1200 K, and (**f**) 1500 K after five ps of the Molecular Dynamic simulations. (**g**) the total energy variations of the B_6_N_6_H_6_ monolayer during the MD simulations at T = 300, 500, 1000, 1200, and 1500 K, Reprinted with permission from Ref. [201].

**Figure 16 biosensors-12-00467-f016:**
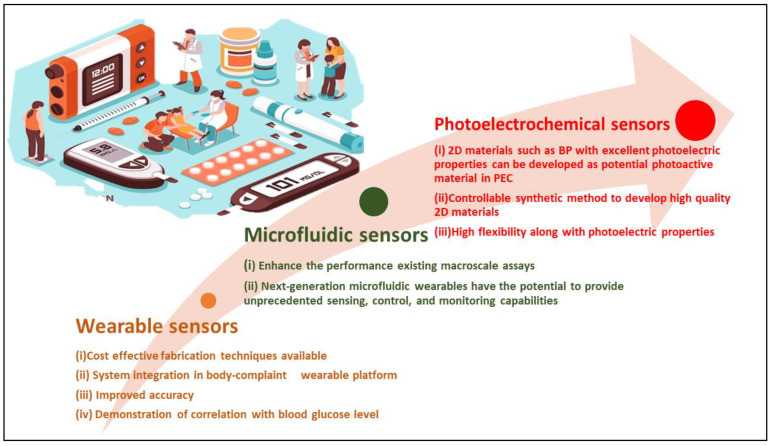
Future prospects on electrochemical glucose sensors based on 2D material.

**Table 1 biosensors-12-00467-t001:** Advantage and disadvantages of electrochemical glucose sensors based on 2D materials.

Pros	Cons
Low power requirements, linear output, and good resolution Excellent repeatability and accuracy Less expensive Fast response time with high sensitivity and low detection limit Multianalyte detection [11]	Narrow or limited temperature range Short or limited shelf life

**Table 2 biosensors-12-00467-t002:** Different types of electrochemical sensors.

Amperometry	Conductometry	Voltammetry	Potentiometry
Amperometry means the measurement of the current flow in a closed loop of cells using an excitation signal produced by the generator Advantages are Good sensitivity and low detection limit Excellent stability makes them particularly suitable for long-term monitoring Disadvantages are Upon applying a high polarizing voltage (Eapp = 0.6–0.8 V), interfering molecules such as uric acid and ascorbic acid, which are commonly present in biological fluids, are also oxidized, leading to nonspecific signals [12]	Response of the conductometric enzyme biosensors is mainly due to protons generated by a biocatalytic reaction inside the layer of immobilized enzyme Advantages are Suitable for miniaturization No light sensitivity Do not need any reference electrode, and the driving voltage can be sufficiently small to decrease the sensor power consumption substantially and reduce safety problems when used in living organisms. Disadvantages are Not very specific (less selective) [13]	Here measurements are related to the recording of either the current–time or the current–voltage relationship by applying known potential varying between the WE and REF electrodes Advantages are Simplicity, sensitivity, speed, and low costs	In this technique, negligible bias current flows as the potential between a working electrode and a reference electrode is measured across some interface Advantages are No extra potential is required Simple, compact The negligible current flow means the technique should be more resistant to interferent effects and ohmic drop considerations Miniaturization without loss of sensitivity Disadvantages are Not very specific (less selective) [14]

**Table 3 biosensors-12-00467-t003:** Properties of 2D materials required for sensing.

2D Materials	Properties Required for Sensing
Graphene	(1) Large surface area aids in the adsorption and diffusion of glucose over the electrode, a larger sensing signal and improved sensitivity. (2) GO as well as rGO can improve the interfacial contact between the electroactive material and the current collector and thereby preventing glucose oxidation (3) Because of the excellent mechanical and thermal stability, graphene can help improve catalyst stability. (4) Biocompatible (5) With its negative charge, hydrophilicity, and surface smoothness, graphene is a good antifouling material that can repel negatively charged and hydrophilic foulants and effectively avoid sensitivity loss.
MXene	(1) Compared to other 2D materials, MXene offers a distinct advantage in terms of electrical conductivity, which is crucial for increasing the rate of heterogeneous electron transport. Its high conductivity is one of the primary reasons for its application in electrochemical glucose sensors. (2) MXene is an easy to synthesize compound with good stability and solution dispersibility. This is crucial for producing electrochemical glucose sensors since drop-casting is the most prevalent process for making modified electrodes, which needs the preparation of the appropriately dispersed coating solution in advance. (3) MXene is an effective substrate material for printing inks. (4) Because of its outstanding stretchability and biocompatibility, MXene can be a powerful substrate for the development of flexible conductive platforms that can be used to construct wearable electrochemical glucose sensors [84].
TMDs	(1) With TMDs nanosheets and their noble metal nanocomposites, electrochemical sensors have improved capabilities due to lower charge transfer resistance and the availability of more reaction sites. (2) A higher surface-to-volume ratio and a smaller size [123] (3) Two-dimensional TMD nanosheets are appealing as electrochemical glucose sensors because of their superior conductivity, huge surface area, high signal/noise ratio and quick electron transfer kinetics and, most importantly, their ability to create composites [124].
LDH	(1) Methods of synthesis are simple and inexpensive (2) Rough surface enhances the mobility of analyte (3) Layers can be effectively tailored to intercalate preferred anions (4) Provide surfaces where enzymes can maintain their activity (5) Allow for easy loading of catalysts (6) Surfaces can be used to immobilize some modifiers or biomolecules.

**Table 4 biosensors-12-00467-t004:** Overview of electrochemical glucose sensors based on two-dimensional materials for glucose detection.

Electrode	Analayte	Linear Range	Stability with Response Current	Detection Limit		Ref
Ni–MoS_2_	Glucose	0–2.4 mM	96.6% after 4 weeks	0.31 μ M		[50]
Cu-nanoflower@AuNPs-GO NFs	Glucose	0.001–0.1 mM	91% after 20 days	0.018 μM		[67]
GOx/rGO/ZnO/Au/PET	Glucose	0.1 to 12 mM	90% after 10 scan cycle	301.9 μM		[68]
Au/CG/C@MWCNTs/PtNPs/GOx/nafion	Glucose	0.5 to 13.5 mM	91.5% after 4 weeks	1.3 μM		[69]
Graphene (G)/platinum oxide (PtO)/n-silicon (Si) heterostructure	Glucose	2–20 mM	-	Sensitivity around 30 μA/mM.cm2		[70]
Graphene paper/AuNPs laser dewetted	Glucose and fructose detection	20 μM–8 mM	-	2.5 μM		[71]
GS/GNR/Ni	Glucose	5 nM–5 mM	91.3% after 30 days	2.5 nM		[72]
Cu/Ni-EG/pNi	Human blood serum	0.0005–1.0 mM	92% after 8 days	6161 mA/mM^−1^ cm^−2^		[73]
NiFe/GO	Glucose	0.05 to 5 mM	Recovery value close to 100%	9 μM		[74]
Pt/Ni@rGO	Glucose	0.02–5.0 mM	98% after 10 weeks	6.3 μM		[75]
rGO/CuSNFs/GCE	Glucose	1 to 2000 µM	97% after 30 days	0.19 µM		[76]
Gra-PANI-co-PDPA-ME	Glucose	1 to 10 µM	94% after 20 days	0.1 µM		[79]
Ag-PANI/rGO	Organic fluids (orange juice, apple juice, mango juice, Coke, and milk)	50 μM to 0.1 μM	91.3 after 30 days	0.79 μM		[80]
Au–PEDOT–ERGO	Glucose	0.1–100 mM	90% after 15 days	0.12 μM		[81]
GOx/Au/MXene/Nafion/GCE	Glucose	0.1 to 18 mM	-	5.9 μM		[87]
Ti_3_C_2_T_x_ MXene–graphene (MG) hybrid	PBS	-	-	0.10 mM in PBS and 0.13 mM in O2- saturated PBS		[88]
Ti_3_C_2_/poly-L-lysine (PLL)/glucose oxidase (GOx)	Glucose	4.0–20 µM and 0.02–1.1 mM	85% after 100 cycles	2.6 μM		[89]
MXene/Nickel–Cobalt layered double hydroxide (NiCo-LDH)	Glucose	0.002–4.096 mM	92.5% after 15 days	0.53 μM		[91]
MXene–Cu_2_O	Glucose	0.01–30 mM	-	2.83 μM		[90]
GOx/3D MGA/GC	Glucose	2 to 20mM	-	0.29 mM		[94]
Cu–MoS_2_ hybrid	Glucose	Up to 4 mM	-		sensitivity 1055 µA mM^−1^ cm^−2^	[95]
Ni–MoS_2_/rGO	Glucose	0.005–8.2 mM	90% after 5 days	2.7 μM		[97]
Ni(OH)_2_/MoSx	Glucose	10–1300 μM	90% after 2 weeks	5.8 μM		[98]
Cu_2_O–MoS_2_/GCE	Glucose	0.01 to 4.0 mM	90% after 10 weeks	1.0 μM		[99]
MoSe_2_/NiO nanorod	Blood and serum	50 µM to 15.5 mM	91%after 50 cycles.	0.6 µM		[101]
Ni_3_Te_2_	Glucose	0.1–0.5 µM	-	25 nM		[103]
3D CoTe_2_	Glucose	-	96% after one month	0.59 μM		[104]
Zn–Cr–ABTS LDH	Glucose	1.3 10^−8^ –6.3 10^−7^ M	-	10 nm		[106]
LDHs/MeOHFc/GOD	Glucose	6.7 × 10^−6^ to 3.86 × 10^−4^ M	-	2.25 mM		[107]
LDHs/CHT/GOD	Glucose	1 × 10^−6^ to 3 × 10^−3^ M	70% after 60 days	0.1 µM		[109]
NiFe–LDH/NF hybrid	Blood and serum	Up to 0.8 mM	-	0.59 μM		[113]
NiAl–LDH/CMC	Glucose	0.2–18.6 mmol L^−1^	-	0.12 mmol L^−1^		[115]
Au/LDH-CNTs-G	Glucose	10 μM to 6.1 mM	95% after 30 days	1.0 μM		[116]
NiCo NSs/GNR-GCE	Glucose	5 μM–0.8 mM 1–10 mM	93% after 3 weeks	0.6 μM		[118]
NiCo-LDH/CCCH/CuF	Glucose	0.001–1.5 mM	-	0.68 μM		[114]
Cu(OH)_2_@CoNi-LDH NT-NSs/GSPE	Glucose	0.002–3.2 3.2–7.7	-	0.6 µM		[119]
VCo-Co(OH)_2_	Glucose	0.4 μM–8.23 mM	-	295 nM		[120]
2D CuO nanoribbon	Glucose	Up to 2 mM	Similar current response after 30 days	58 µM		[121]
**Wearable electrochemical sensors**						
Ag/Pt @rGO	Sweat			48 μA/mMcm2		[128]
PtCo/NPG/GP	Blood	35 μM–30 mM	88% after 30 days	5 μM		[125]
GOx/PtNP/acetic acid-treated LIG	Sweat	2.1 mM	-	0.3 µM		[135]
CNTs/Ti_3_C_2_T_x_/PB/CFMs	Sweat	10 × 10^−6^–22 × 10−3 M	-	Glucose	Lactate	[136]
				0.33 × 10^−6^ M	0.67 × 10-6 M	
rGO-TEPA/PB	Human sweat and blood	0.1–25 mM	75.1% after 2 weeks	25 μM		[130]
Ti_3_C_2_Tx/MB	Sweat	0.08–1.25 mM	-	Glucose	Lactate	[137]
				17.05 μM	3.73 μM	
ZnO TPs/MXene	Sweat	0.05–0.7 mM	-	17.05 μM		[138]
GOx/gold/MoS_2_/gold	Glucose	500–100 nm	-	10 nM		[139]
NiCo LDH/CC	Glucose	1 μM to 1.5 mM	95% after one month	0.12 μM		[142]
NiSe_2_	Glucose	0.1–1 mM	-	24.8 μM		[140]
NiAl-LDH	Glucose	1 to 329 µM	94.7% after one month	0.22 µM		[143]
**Photoelectrochemical sensor**						
Graphene–CdS hybrid	Glucose	0.1∼4mmol dm^−3^	-	7 μmol dm-3		[152]
CoOx/graphene-CdS/GCE	Urine	5–370 μM	-	0.5 μM		[153]
rGO-CdS QDs/PBA/P-NB/GDH	Glucose	Upto 200 μM	95% response to NADH oxidation	<1mM		[154]
LI-NiEC-CdS-G@ITO	Glucose	-	-	0.4 μM		[155]
Cu_2_O-rGO TiO_2_ NTs/Ti electrode	Glucose	0.0007–20 mM	Photocurrent response retained upto 94% after 7 weeks	0.21 μM		[156]
rGO/TiO_2_ nanotube	Glucose	-	-	5 μM		[157]
BiOCl-G NHS	Human serum	500 µM–10mM	97% after 4 weeks	0.22 mM		[158]
BiOBr-TNTA	Human serum	5 × 10^−2^–3 × 10^7^ nM	95% after 4 weeks	10 nM		[159]
g-C_3_N_4_/ZnIn_2_S_4_	Glucose	1–10,000 μM		0.28 μM		[160]
g-C_3_N_4_/WO_3_	Glucose	0.01~7.12 mM	89.5% after 30 days	0.1 μM		[162]
g-C_3_N_4_/Fe_2_O_3_	Glucose	g 0.1 to 11.5 mg L^−1^	-	0.03 mg/L		[161]
ITO/TiO_2_-Au NPs- g-C3N4 -MnO_2_	Glucose and lactose	0.004–1.75 mM	92.3% after 5 weeks	0.12 µM		[163]
Ti_3_C_2_ MXene/Cu_2_O	Human serum samples	-	-	0.17 nM		[164]
TiO_2_/Ti_3_C_2_/Cu_2_O	Human serum samples	-	-	33.75 nM		[165]
MoS_2_ ultrathin nanosheets	Glucose	-	98%	0.61 nmol/L		[166]
**Other glucose sensors**						
rGO-PtNW	Glucose	0.032–1.89 mmol/L	-	4.6 μmol/L		[171]
GOx/Co_3_Mn-CO_3_/CF	Glucose	-	-	0.02 mM		[172]
NiCo_2_N/N-doped graphene	Glucose	2.008 µM to 7.15 mM	98.52%	0.05 µM		[174]
Ni_3_N nanosheets	Glucose	0.2 μM to 1.5 mM	91.3% after one month	0.06 µM		[175]
Ni_3_N/NCS	Glucose	1 μM–3000 μM and 3000 μM–7000 μM	-	0.1 μM and 0.35 μM		[176]

**Table 5 biosensors-12-00467-t005:** Binding energy, bond distances, and charge transfer of various glucose adsorption systems.

System	Adsorption Energy (Ev)	Bond Distance (Å)	Charge Transfer ∆*q* (E)	Ref.
Graphene + Au + glucose	−0.71	Au—O = 2.28	−0.21	[194]
Graphene + Pt + glucose	−0.38	Pt—O = 2.39	−0.17	[194]
C_2_N + glucose	−0.93 (gas phase) −1.32 (aq. phase)		−0.01	[195]
HGr + Cl + glucose	−1.86	Cl—O = 2.850 H—C = 1.746	−0.20	[184]
HGr + N + glucose	−1.69	N—O = 3.372 H—C = 2.642	−0.08	[184]
HGr + O + glucose	−1.73	O—O = 2.926 H—C = 2.371	−0.12	[184]

## Data Availability

Not applicable.

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
