# Peer review of "Recent Developments and Future Perspective on Electrochemical Glucose Sensors Based on 2D Materials"

_biosensors, 2022, doi:10.3390/bios12070467_

Round 1

Reviewer 1 Report

The authors performed a deep and extensive study of the glucose-sensing systems. The information is important for researchers that want to study in this specific area, though the review is quite long. It will be nice to be more compact.

Few minor English checks are required, such as Page 2 Line 55. Sentence does not make sense. Please rephrase.

Also alongside with reference 9-11, which are old papers, the following one DOI: 10.1149/1945-7111/abe8b6, covers those aspects and is a paper from 2021. The authors may consider adding it to the reference list.

Reviewer 2 Report

The authors have well described a good state of art in the current research of glucose biosensors, covering the non- and enzymatic ones with a great examples reported in the literature. Unfortunately, I consider that must doble check some english and structure errors in the manuscript (i.e. Line 47, 522,) use of possesive for objects "Given graphene´s excelent properties" (i.e. line 579). 

There are some problems with the unit, in some cases is microA/mM-1 (which is definitively an error), or 75 A mmol-1 L cm-2. Please check and give a consistent units. Check the line 735 as example, LOD is mentioned previous to its meaning as limit of detection (LOD). Please check the writting of the manuscript.

I would like to suggest:

- The fundamental studies performed by Feliu´s group in oxidation of glucose in Pt electrodes: https://doi.org/10.1016/j.electacta.2020.136765, https://doi.org/10.1016/j.jelechem.2020.114549

-Also the following references, related with enzymatic electrode materials for glucose sensing using direct electron system, entrapment and new metal-free catalyst:

https://doi.org/10.1016/j.electacta.2021.138530

https://doi.org/10.1016/j.bios.2017.01.058

https://doi.org/10.1016/j.bioelechem.2020.107487

https://doi.org/10.1016/j.electacta.2020.137434

https://doi.org/10.1016/j.bioelechem.2015.04.005

https://doi.org/10.1016/j.talanta.2021.122386

https://doi.org/10.1038/s41467-018-06106-3

Reviewer 3 Report

Ref: biosensors-1745875

Title of the manuscript: “Recent Developments and Future Perspective on Electrochemical Glucose Sensors based on 2D materials.”

In this paper, Sithara et al. complied with the recent developments on 2D materials as Electrochemical glucose sensors. The manuscript is nicely written and has many figures for easy understanding for beginners only. But the article is quite lengthy, monotonous, and needs to be refined and crisp. Moreover, the article revolves around the mechanism of glucose sensors for the first ten pages which is too sluggish. This manuscript can be recommended for publication after minor revision.

  1. Authors can redefine the manuscript headings. There is overlapping of all the headings which doesn’t help in distinguishing point by point.
  2. Just adding too many references doesn’t serve the purpose. A critical analysis of the reported literature is required in terms of detection procedures reported. More points regarding the performance, stability, and robustness shall be incorporated.
  3. Authors can include the market statistics about the enzymatic and non-enzymatic based glucose sensors with regard to 2D materials.
  4. There is no Table caption for Table 2. Moreover, the electrodes should be numbered. There should be a uniformity in electrode type used. The format for Table 2 is not the same throughout. Figures reprinted should be of high resolution.

Reviewer 4 Report

This paper gives a overview of the recent developments and future perspective on electrochemical glucose sensors based on 2D materials. The article provides a very detailed review of the enzymatic and non-enzymatic glucose sensor applications. The work is well presented however the reviewer has some suggestions, which are outlined below. Before the acceptance of the article, the authors need to make the following corrections:

1.      The authors have not mentioned about the LoQ of the different Sensors that they have reviewed. The LoQ values must also be included in whichever studies have mentioned about it.

2.      An SROC curve with sensitivity and specificity values of the selected studies could enhance the quality of the article.

3.      The objective of the paper in term of selecting Electrochemical Glucose Sensors when compared with other methods is not clear. The authors should address it in clear way by comparing various other methods and proving advantages or disadvantages of the methods and

4.      Even though the review is about glucose sensors based on electrochemical measurements, the authors must clearly state the advantages over the other methods such as Impedance etc.

5.      Extensive editing of English language and style is required.

6.      There are still some places that have not been carefully explained. For example, some related applications with their method. For a balanced overview and in order to provide more motivations for the present work, the authors should include some studies of the latest examples of nanorod structure, in the introduction section. 

Suggested reference:

R1: I-Chen Wu et al.” Nanostructure ZnO/Cu2O Photoelectrochemical and Self-powered Biosensor for Esophageal Cancer Cell Detection,” Optics Express 25(7), 7689-7706 (2017).

Round 2

Reviewer 4 Report

The author has fully responded to my question. This article can be accepted by Biosensors.